

# New insights into aerosol and climate in the Arctic

Jonathan P.D. Abbatt[1], W. Richard Leaitch[2], Amir A. Aliabadi[3], Allan K. Bertram[4], Jean-Pierre Blanchet[5], Aude Boivin-Rioux[6], Heiko Bozem[7], Julia Burkart[8], Rachel. Y. W. Chang[9], Joannie Charette[6], Jai. P. Chaubey[9], Robert J. Christensen[1], Ana Cirisan[5], Douglas B. Collins[10], Betty Croft[9], Joelle Dionne[9], Greg J. Evans[11], Christopher G. Fletcher[12], Roya Ghahreman[2], Eric Girard[5,*], Wanmin Gong[2], Michel Gosselin[6], Margaux Gourdal[13], Sarah J. Hanna[4], Hakase Hayashida[14], Andreas B. Herber[15], Sareh Hesaraki[16], Peter Hoor[7], Lin Huang[2], Rachel Hussherr[13], Victoria E. Irish[4], Setigui A. Keita[5], John K. Kodros[17], Franziska Köllner[7,18], Felicia Kolonjari[2], Daniel Kunkel[7], Luis A. Ladino[19], Kathy Law[20], Maurice Levasseur[13], Quentin Libois[5], John Liggio[2], Martine Lizotte[13], Katrina M. Macdonald[11], Rashed Mahmood[14,21], Randall V. Martin[9], Ryan H. Mason[4], Lisa A. Miller[22], Alexander Moravek[1], Eric Mortenson[14], Emma L. Mungall[1], Jennifer G. Murphy[1], Maryam Namazi[23], Ann-Lise Norman[24], Norman T. O'Neill[16], Jeffrey R. Pierce[17], Lynn M. Russell[25], Johannes Schneider[18], Hannes Schulz[15], Sangeeta Sharma[2], Meng Si[4], Ralf M. Staebler[2], Nadja S. Steiner[22], Martí Galí[13], Jennie L. Thomas[20], Knut von Salzen[21], Jeremy J.B. Wentzell[2], Megan D. Willis[26], Gregory R. Wentworth[1,27], Jun-Wei Xu[9], Jacqueline D. Yakobi-Hancock[28]

[1]Department of Chemistry, University of Toronto, Toronto, Canada
[2]Environment and Climate Change Canada, Toronto, Canada
[3]School of Engineering, University of Guelph, Guelph, Canada
[4]Department of Chemistry, University of British Columbia, Vancouver, Canada
[5]Department of Earth and Atmospheric Sciences, Université du Québec à Montréal, Montréal, Canada
[6]Institut des sciences de la mer de Rimouski, Université du Québec à Rimouski, Rimouski, Canada
[7]Institute for Atmospheric Physics, Johannes Gutenberg University, Mainz, Germany
[8]Aerosol Physics & Environmental Physics, University of Vienna, Vienna, Austria
[9]Department of Physics and Atmospheric Science, Dalhousie University, Halifax, Canada
[10]Department of Chemistry, Bucknell University, Lewisburg, USA
[11]Department of Chemical Engineering and Applied Chemistry, University of Toronto, Toronto, Canada
[12]Department of Geography and Environmental Management, University of Waterloo, Waterloo, Canada.
[13]Department of Biology, Université Laval, Quebec City, Canada
[14]School of Earth and Ocean Sciences, University of Victoria, Victoria, Canada
[15]Alfred Wegener Institute, Helmholtz Center for Polar and Marine Research, Bremerhaven, Germany
[16]Centre d'Applications et de Recherches en Télédétection, Université de Sherbrooke, Sherbrooke, Canada
[17]Department of Atmospheric Science, Colorado State University, Fort Collins, USA
[18]Particle Chemistry Department, Max Planck Institute for Chemistry, Mainz, Germany
[19]Centro de Ciencias de la Atmósfera, Universidad Nacional Autónoma de México, Ciudad Universitaria, México City, México
[20]ATMOS/IPSL, Sorbonne Université, UVSQ, CNRS, Paris, France



[21]Canadian Centre for Climate Modelling and Analysis, Environment and Climate Change Canada, Victoria, Canada
[22]Institute of Ocean Sciences, Fisheries and Oceans Canada, Sidney, Canada
[23]Department of Mathematics, University of Isfahan, Isfahan, Iran
[24]Department of Physics and Astronomy, University of Calgary, Calgary, Canada
[25]Scripps Institution of Oceanography, University of California, San Diego, La Jolla, USA.
[26]Lawrence Berkeley National Laboratory, Berkeley, USA
[27]Alberta Environment and Parks, Edmonton, Canada
[28]National Research Council, Ottawa, Canada

*Correspondence to*: Jonathan P.D. Abbatt (jabbatt@chem.utoronto.ca) or W. Richard Leaitch (richard.leaitch@canada.ca)

*\*This paper is dedicated to Eric Girard, a NETCARE scientist who died July 10, 2018. Eric contributed greatly to the field of Arctic cloud and aerosol microphysics during his research career.*

**Abstract.** Motivated by the need to predict how the Arctic atmosphere will change in a warming world, this article summarizes recent advances made by the research consortium NETCARE (Network on Climate and Aerosols: Addressing Key Uncertainties in Remote Canadian Environments) that contribute to our fundamental understanding of Arctic aerosol particles as they relate to climate forcing. The overall goal of NETCARE research has been to use an interdisciplinary approach encompassing extensive field observations and a range of chemical transport, earth system, and biogeochemical models. Several major findings and advances have emerged from NETCARE since its formation in 2013. 1) Unexpectedly high summertime dimethyl sulfide (DMS) levels were identified in ocean water and the overlying atmosphere in the Canadian Arctic Archipelago (CAA). Furthermore, melt ponds, which are widely prevalent, were identified as an important DMS source. 2) Evidence was found of widespread particle nucleation and growth in the marine boundary layer in the CAA in the summertime. DMS-oxidation-driven nucleation is facilitated by the presence of atmospheric ammonia arising from sea bird colony emissions, and potentially also from coastal regions, tundra, and biomass burning. Via accumulation of secondary organic material (SOA), a significant fraction of the new particles grow to sizes that are active in cloud droplet formation. Although the gaseous precursors to Arctic marine SOA remain poorly defined, the measured levels of common continental SOA precursors (isoprene and monoterpenes) were low, whereas elevated mixing ratios of oxygenated volatile organic compounds were inferred to arise via processes involving the sea surface microlayer. 3) The variability in the vertical distribution of black carbon (BC) under both springtime Arctic haze and more pristine summertime aerosol conditions was observed. Measured particle size distributions and mixing states were used to constrain, for the first time, calculations of aerosol-climate interactions under Arctic conditions. Aircraft- and ground-based measurements were used to better establish the BC source regions that supply the Arctic via long-range transport mechanisms. 4) Measurements of ice nucleating particles (INPs) in the Arctic indicate that a major source of these particles is mineral dust, likely derived from local sources in the summer and long-range transport in the spring. In addition, INPs are abundant in the sea surface microlayer in the Arctic, and





possibly play a role in ice nucleation in the atmosphere when mineral dust concentrations are low. 5) Amongst multiple aerosol components, BC was observed to have the smallest effective deposition velocities to high Arctic snow.

# 1 Introduction

Rapid changes in the Arctic environment including rising temperatures, melting sea ice, elongated warm seasons and changing long-range transport patterns (IPCC, 2013) are driving a growing interest in developing a better understanding of the processes that control Arctic climate. Furthermore, because high-latitude climate change is a bellwether for change on a global scale, it is particularly important to understand the processes that lead to Arctic amplification of radiative forcing (Serreze and Barry, 2011).

This article discusses key discoveries that have been made in climate-related Arctic aerosol research by the NETCARE (Network on Climate and Aerosols: Addressing Key Uncertainties in Remote Canadian Environments) research network. Formed in 2013, NETCARE consists of Canadian academic and government researchers along with international collaborators. Given the highly diverse nature of inter-related earth system processes that couple within the Arctic environment, the network is necessarily interdisciplinary, consisting of climate and air quality modellers, atmospheric chemists, aerosol and cloud physicists, biological and chemical oceanographers, biogeochemists, and remote sensing experts. Over the past six years, the network has conducted a set of field campaigns and modelling projects focused on the sources and loss mechanisms of atmospheric particles, their chemical and optical characteristics, and their role in climate. The field studies were conducted using a variety of platforms including the Alfred Wegener Institute's Polar 6 aircraft (Herber et al., 2008), the research icebreaker Canadian Coast Guard Ship (CCGS) *Amundsen*, and the Dr. Neil Trivett Global Atmosphere Watch Observatory at Alert, Nunavut (hereafter, Alert). Table 1 and Fig. 1 present the locations and dates of the field studies. The modelling studies used the Canadian Atmospheric Global Climate Model CanAM (von Salzen et al., 2013), the GEOS-Chem chemical transport model with associated microphysics module TOMAS (Croft et al., 2016b), Environment and Climate Change Canada's GEM-MACH chemical transport model (Moran et al., 2010), a biogeochemical model coupling the ocean-sea ice-atmosphere, coupled ice-ocean-biogeochemistry models in 1-D and 3D configurations (Hayashida et al., 2018a; Mortenson et al., 2018), and the Lagrangian particle dispersion model FLEXPART (Stohl et al., 2005). The overall goals of the network have been to study the nature of fundamental biogeochemical and physical processes that connect aerosol to climate in environments that vary from pristine to polluted, such as those found in the Arctic, in order to use this new understanding to improve the accuracy of the different modelling approaches used to simulate climate in these environments.

The network's output is documented through a special issue across three journals, *Atmospheric Chemistry and Physics; Biogeosciences;* and *Atmospheric Measurement Techniques* (https://www.atmos-chem-phys.net/special_issue835.html), of which this article is a part. NETCARE has also produced a number of publications in other journals. The specific goal of this overview paper is to synthesize the results from NETCARE and to act as a gateway into the more detailed results described within the special issue and elsewhere.



Written for a scientist interested in the fields of Arctic climate, atmospheric chemistry and biogeochemistry, this article starts with a background on Arctic aerosol that is not focused on NETCARE results (Section 2). For additional background information, the reader is referred to (Quinn et al., 2006, 2008) and (Law and Stohl, 2007). The article then presents new insights into the three topics around which NETCARE was structured: marine processes and the Arctic atmosphere (Section 3), the sources, sinks and properties of Arctic aerosol (Section 4), and ice nucleating particles (INPs) (Section 5). Each of these sections stands alone, so that the interested reader can focus their attention on a specific subject. However, there are clear connections between the different topics. For example, Section 3 (Marine processes and the Arctic atmosphere) is motivated by the increasing marine impact that is arising as sea ice melts and focuses on new NETCARE Arctic measurements of dimethyl sulfide (DMS), ammonia and oxygenated volatile organic carbon species. The oceans are an important source of such reactive gases to the atmosphere, leading to direct impacts on aerosol particles and ultimately on climate. Those connections are made in Section 4 (Arctic aerosol: sources, sinks and properties), which presents insights gleaned for the summertime environment, when these marine emissions can lead to new particle formation and growth, and discusses the impacts of this aerosol on clouds. Section 4 also presents results from the Arctic haze springtime period, where the emphasis is on the sources of particles, their optical properties and the potential for direct radiative forcing. Section 5 (Ice nucleating particles) addresses the select fraction of atmospheric particles that nucleate ice crystals. Section 6 concludes the article by discussing remaining research uncertainties and future priorities.

## 2 Background on Arctic aerosol

Over the last half century, our knowledge of Arctic aerosol and its role in climate has advanced from almost nothing to a clear understanding of its importance, although important questions remain regarding mechanistic details. This short section of the paper presents a comprehensive description of the field, leaving the recent NETCARE results for later sections.

Following early observations of visibility-reducing haze particles in the spring Arctic atmosphere (Greenaway, 1950), study of Arctic haze began in earnest in the 1970s (Holmgren et al., 1974; Rahn and Heidam, 1981). Investigations intensified through the 1980s, with observations (ground-based and airborne) and meteorological analyses indicating that haze particles were transported from mid-latitude pollution sources, often in layers that reached up to the tropopause, and that their concentrations increased in winter and spring due to efficient meridional transport and low rates of wet deposition (Barrie, 1986; Barrie and Hoff, 1985; Brock et al., 1989; Leaitch et al., 1989; Radke et al., 1984; Schnell and Raatz, 1984; Shaw, 1982). Through the 1990s and beyond, concentrations of Arctic haze components declined at the northernmost observatories: Alert, Nunavut; Barrow, Alaska; Mount Zeppelin, Svalbard; and Station Nord, Greenland (Heidam et al., 1999; Hirdman et al., 2010; Quinn et al., 2009; Sharma et al., 2004, 2006; Sinha et al., 2017; Sirois and Barrie, 1999). Recent measurements (Fisher et al., 2011; Frossard, 2011; Leaitch et al., 2018b; Massling et al., 2015; Sharma et al., 2017; Sinha et al., 2017) have found surface mass concentrations of sulfate, organic material and black carbon (BC) 3–10 times lower than those estimated from studies conducted prior to 1981 (Rahn and Heidam, 1981), but the total Arctic column burden of BC may have increased (Koch and





Hansen, 2005; Sharma et al., 2013) with implications for climate forcing efficiency (Breider et al., 2017). The turn of the century saw renewed interest in Arctic haze with concern for the role of BC in Arctic warming (Flanner et al., 2007; Hansen and Nazarenko, 2004; Law and Stohl, 2007; McConnell et al., 2007; Quinn et al., 2008; Shindell and Faluvegi, 2009).

From the early studies of Arctic haze arose the concept of the Arctic atmosphere as a dome of cold air that regulates transport of polluted air from southerly latitudes (Barrie, 1986). The polar front extends in the winter to include more southerly industrial emissions that can be transported into the high Arctic, and the front retreats in the summer to inhibit transport from mid-latitude sources. Figure 2 shows an example of identification of the polar dome in spring 2015 through measurements conducted during the NETCARE aircraft campaign. Pollution transport into the Arctic may also be influenced by the North Atlantic Oscillation (Duncan and Bey, 2004; Eckhardt et al., 2003). Arctic haze originates from Eurasia, Siberia, southeast Asia and North America, with Eurasia as the dominant source region at lower altitudes and contributions from south/central Asian sources dominating at higher altitudes (Fisher et al., 2011; Qiu et al., 2017; Sharma et al., 2013; Stohl, 2006). Sea salt contributes to the haze due to the combination of stronger winds and reduced wet deposition in the winter and spring (Huang and Jaeglé, 2017; Leaitch et al., 2018b) and frost flowers may contribute some marine salt (Shaw et al., 2010). Snowpack exchange is a potential springtime source of organic precursors (McNeill et al., 2012), while stratospheric contributions appear to be small (Leaitch and Isaac, 1991; Stohl, 2006).

Arctic haze warms the Arctic in several ways. BC from anthropogenic sources and forest fires deposits to snow and ice, lowering the surface albedo (Clarke and Noone, 1985; Doherty et al., 2010; Flanner et al., 2007; Forsström et al., 2013; Hegg and Baker, 2009; Keegan et al., 2014; McConnell et al., 2007). Atmospheric haze layers containing BC are warmed while the underlying surface is cooled, which acts to increase atmospheric stability (Blanchet and List, 1983; Brock et al., 2011; Koch and Genio, 2010; Leighton, 1983; Pueschel and Kinne, 1995; Valero et al., 1984). Meridional temperature gradients are enhanced by BC outside the Arctic, which warms the air during transport to the Arctic, hence increasing heat transport into the Arctic (Sand et al., 2013). Dust, when present in layers over high albedo surfaces and/or deposited to the snow, will warm the atmosphere (Bond et al., 2013; Dumont et al., 2014; Lohmann and Feichter, 2005). Arctic haze can also increase longwave radiative forcing by forming thin Arctic low-level liquid clouds (Garrett et al., 2009; Garrett and Zhao, 2006; Lubin and Vogelmann, 2006; Mauritsen et al., 2011).

However, many components of Arctic haze (e.g., sulfate, OM, sea salt) help to cool the Arctic by scattering light back to space (Schmeisser et al., 2018) and by modifying the microphysics of liquid clouds to enhance shortwave cooling (Garrett and Zhao, 2006; Lubin and Vogelmann, 2006; Zamora et al., 2017; Zhao and Garrett, 2015). During winter and spring, sulfuric acid in Arctic haze particles may reduce their effectiveness as INPs, leading to larger crystals that precipitate more easily. As a result, there may be an increase in the dehydration rate of the atmosphere and a corresponding reduction in longwave forcing (Blanchet and Girard, 1994; Curry and Herman, 1985). At cirrus temperatures, dust, ammonium sulfate and sea salt may also increase cloud albedo by increasing ice crystal concentrations (Abbatt et al., 2006; Sassen et al., 2003; Wagner et al., 2018). Observed and simulated seasonal cycles of BC and sulfate typically show a maximum in near-surface concentrations in March or April (Barrie and Hoff, 1985; Eckhardt et al., 2015; Garrett et al., 2010; Sharma et al., 2006) and clean conditions in the




summertime. Natural emissions of BC from vegetation fires are considerable in late spring to early summer in the Arctic and at mid-latitudes (Mahmood et al., 2016). Production of sulfate aerosol is more efficient in the warm than the cold seasons (Mahmood et al., 2018; Tesdal et al., 2015). Thus, the decline in Arctic haze after its peak in early spring and the approach to the summertime pristine conditions are largely related to aerosol scavenging rather than a reduction in aerosol production. Wet

deposition associated with transport across the retracted polar front and frequent low-intensity precipitation within the polar dome keeps the summertime Arctic nearly free of anthropogenic aerosol (Barrie, 1986; Browse et al., 2012; Garrett et al., 2010) while marine sources have a strong influence on the Arctic summer aerosol (Dall´Osto et al., 2017; Korhonen et al., 2008b; Stohl, 2006).

Summer sources of sulfate appear to be the oxidation of DMS from the Arctic Ocean as well as connected waters to the south,

volcanism, residual Arctic haze sulfate and some anthropogenic sulfate or $SO_2$ that may leak past the Arctic front into the dome (Leaitch et al., 2013). Methane sulfonic acid (MSA), another product of DMS oxidation, is most prominent in the spring and summer, and its levels are linked to the northward migration of the marginal ice zone (Laing et al., 2013; Leck and Persson, 1996; Quinn et al., 2009; Sharma et al., 2012). Aside from DMS, natural sources that can contribute to summertime Arctic atmospheric organic matter include biomass burning (Chang et al., 2011a; Stohl, 2006) and sea spray (Chang et al., 2011a;

Frossard et al., 2014; Shaw et al., 2010). Sea spray encompasses marine emissions of aerosol precursors, products of photochemical processes transforming organic compounds at the ocean surface, and colloidal gels (Leck and Bigg, 1999, 2005, 2007; Orellana et al., 2011).

Characterized by a unimodal diameter distribution centred between 200 and 300 nm (Bigg, 1980; Heintzenberg, 1980; Leaitch and Isaac, 1991; Radke et al., 1984; Staebler et al., 1994), Arctic haze particles are effective at both scattering light (Andrews

et al., 2011; Schmeisser et al., 2018) and acting as nuclei for cloud droplets (Earle et al., 2011; Komppula et al., 2005). In contrast, the summertime number distribution is dominated by smaller Aitken particles resulting from newly formed particles that have experienced modest growth in the near-pristine summer Arctic. Their small sizes render Aitken particles relatively ineffective at scattering light, but they may be able to influence cloud microphysics in the clean summertime Arctic (Korhonen et al., 2008b).

Overall, the net effect of anthropogenic aerosols has been to cool the Arctic (Fyfe et al., 2013; Najafi et al., 2015), and Navarro et al. (2016) showed that reductions in Arctic haze have contributed to the sharp increase in the rate of Arctic warming since 1990. Mitigation of BC emissions may help to slow Arctic warming so long as cooling components are not simultaneously mitigated (Kopp and Mauzerall, 2010; Sand et al., 2013; Shindell and Faluvegi, 2009).

As seen from this brief overview, understanding natural aerosol processes in addition to anthropogenic aerosol sources is vital

for climate studies, as anthropogenic aerosol forcing is measured against the natural component (Carslaw et al., 2013; Megaw and Flyger, 1973). For example, in the winter and spring, sea salt aerosol may play an important climate role (Kirpes et al., 2018). At the start of NETCARE, detailed knowledge of natural particle sources and their impacts on clouds in the nearly pristine summer was incomplete, and it became a major focus of the network's research activities.



## 3 Marine processes and the summertime Arctic atmosphere

### 3.1 Rationale and research questions

In remote marine atmospheres such as the summertime Arctic, assessing the impact of natural marine biogenic aerosol (MBA) sources on cloud formation is pivotal to accurately estimating climate forcing (Carslaw et al., 2013; Charlson et al., 1987). While a variety of organic compounds, such as marine microgels, may be relevant primary MBA sources in the Arctic (Leck and Bigg, 2005; Orellana et al., 2011), DMS-derived sulfate is thought to be a key precursor to secondary marine aerosol mass over biologically productive regions (Ghahremaninezhad et al., 2016; Leaitch et al., 2013; McCoy et al., 2015; Park et al., 2017). The production of DMS and other organic compounds in polar regions is linked to the productivity of microalgae, as well as to the dynamics and the structure of pelagic (oceanic) and sympagic (ice-associated) microbial food webs (Gabric et al., 2017; Levasseur, 2013; Simó, 2001; Stefels et al., 2007). Peaks in the DMS proxy MSA have been observed in association with bursts of phytoplankton productivity in the high Arctic (Becagli et al., 2016). As well, atmospheric DMS mixing ratios in the marine boundary layer have been shown to transiently peak during the phytoplankton growth period from May to September (Park et al., 2013, 2018). Particle nucleation and growth events have been observed even at moderate levels of atmospheric and oceanic DMS in the High Canadian Arctic (Chang et al., 2011b; Rempillo et al., 2011).

Despite these compelling indications of the key role played by marine biogenic DMS in contributing to sulfate aerosols (Rempillo et al., 2011), measurements of seawater and sea-ice DMS during the biologically productive summer months (June to August) that coincide with clean aerosol time periods are still scarce (Jarníková et al., 2018; Levasseur, 2013). The paucity of DMS measurements in ice-associated habitats, such as under the sea ice, in melt ponds atop the ice, or directly at the Arctic sea ice margin, is even greater (Levasseur, 2013). Sea ice not only acts to modulate gaseous exchange but also hosts active microorganisms (Gradinger, 2009), making it a fundamental driver of various MBA precursors, including DMS (Arrigo, 2014; Gabric et al., 2017; Korhonen et al., 2008b). Our understanding of the processes that control other key gases that can lead to aerosol formation in marine environments, including ammonia and volatile organic carbon compounds (VOCs), is particularly weak. There have been very few measurements of their Arctic abundance in the past and we have a poor understanding of their sources. In this context, NETCARE targeted the spatiotemporal variability of DMS and the underlying ecosystemic mechanisms controlling its abundance in the Eastern Canadian Arctic (Canadian Arctic Archipelago, henceforth CAA, and Northern Baffin Bay), along with the atmospheric abundances and sources of other key gases.

### 3.2 DMS production in oceanic and ice-associated environments

The two NETCARE summer campaigns (July–August 2014 and 2016) revealed high open water concentrations of DMS (interquartile range of 5.1–10.9 nmol L$^{-1}$, maximum 75 nmol L$^{-1}$) in the Eastern Canadian Arctic. Previous pan-Arctic measurements had an interquartile range of 0.9–5.9 nmol L$^{-1}$ and a maximum of 26 nmol L$^{-1}$. These results challenged the representativeness of measurements conducted during previous cruises in the Eastern Canadian Arctic in late summer and early fall (Luce et al., 2011; Motard- Côté et al., 2012) by showing that average summer surface DMS concentrations in this



part of the Arctic were at least two-fold higher than measurements conducted later in the season. The range of seawater DMS concentrations measured in the CAA during the NETCARE expeditions in 2014 (Fig. 3) and 2016 is comparable to those observed in the same area and season in 2015 by Jarníková et al. (2018), who found the highest DMS concentrations in association with localized peaks of chlorophyll *a*, a proxy of phytoplankton biomass. Combining oceanic and atmospheric

NETCARE datasets provided further evidence that marine DMS hotspots were associated with high atmospheric DMS (Mungall et al., 2016). As described in Section 4, connections were also found between localized regions of high oceanic biological activity and new particle formation and growth events (Collins et al., 2017; Mungall et al., 2016) that may be partly caused by DMS and organic emissions. These new observations lend strong support to the hypothesis that local Arctic DMS sources are responsible for the summertime peaks in MSA measured at Alert (Leaitch et al., 2013; Sharma et al., 2012).

Novel measurements made during NETCARE also substantiated the potentially important role played by melt ponds. An in-depth study of nine melt ponds revealed that brackish melt ponds over first-year sea ice (FYI) may have DMS concentrations ranging from 3 to 6 nmol L$^{-1}$ (Fig. 3) with an average of 3.7 nmol L$^{-1}$ (Gourdal et al., 2018). These concentrations are comparable to the global oceanic annual average of ca. 4.2 nmol L$^{-1}$ derived from the PMEL database. A search for the underlying mechanisms associated with the presence of DMS in these melt ponds revealed that intrusions of seawater through

permeable sea ice is a key physical process allowing their colonization by DMS-producing marine protists (Gourdal et al., 2018). Considering that the areal coverage of melt ponds may extend up to 90 % over Arctic FYI (Rösel et al., 2012) these results shed light on a previously overlooked source of DMS to the atmosphere and further call for a re-evaluation of the emissions from these regions within climatologies that currently assume the absence of DMS fluxes above ice-covered waters (Lana et al., 2011). In a simulation exercise, melt ponds were found to contribute an average of 20 % (and up to 100 %) of the

atmospheric DMS over and near ice-covered regions of the Arctic during the melt season (Mungall et al., 2016).
While marginal ice zones (MIZ) and various ice-edge systems have long been recognized for their teeming biological activity (Mundy et al., 2009; Perrette et al., 2011) and potential for heightened DMS production (Galí and Simó, 2010; Levasseur, 2013; Matrai and Vernet, 1997), they remain surprisingly under-documented for their specific role in MBA production in the Eastern Canadian Arctic during summer. Two distinct MIZs explored during the summer of 2014 revealed highly contrasting

DMS dynamics, suggesting that whether the sea ice is FYI or multi-year ice (MYI) is of paramount importance in shaping marine food webs and the net production of DMS in the water exiting the ice pack. At the MYI edge in Kennedy Channel (ca. 81° N), DMS concentrations were very low at the ice edge and increased progressively over several kilometres as the water flowed away from the ice pack, suggesting that time out from under the ice was required for development of a phytoplankton bloom and the concomitant production of DMS. However, at a FYI edge in the CAA (ca. 74° N), DMS concentrations were

already high under the ponded sea ice (Fig. 3) due to the presence of an under-ice bloom. Consequently, the surface waters exiting the ponded FYI displayed high DMS even at the very edge of the ice pack. The elevated levels of DMS persisted for several kilometres away from the ice edge. Thus, beyond the direct contributions melt ponds make to DMS fluxes, results from the NETCARE campaigns suggest that melt ponds play an additional role in DMS dynamics by promoting the earlier onset of under-ice phytoplankton blooms and DMS production (Lizotte et al., 2018). Taken together, these observations reveal the



potential for high DMS emissions to the atmosphere immediately upon the cracking, opening or melting of ponded FYI without the prerequisite of an ice-free period to initiate a phytoplankton bloom and potential accumulation of DMS in surface waters.

### 3.3 Gaseous aerosol precursors in Arctic marine and coastal environments

High levels of DMS have previously been associated with aerosol formation and growth in the CAA (Chang et al., 2011b; Park et al., 2017; Rempillo et al., 2011). As part of NETCARE, new atmospheric measurements of DMS were performed from both the Polar 6 aircraft and the CCGS *Amundsen* icebreaker. Mean DMS mixing ratios in the Arctic lower free troposphere in April 2015 were found to be unexpectedly high (average $116\pm8$ ppt) relative to those from the July 2014 campaign ($20\pm6$ ppt) (Ghahremaninezhad et al., 2017). The springtime levels likely reflect long-range transport from more southerly, open ocean regions. In the summertime, the boundary layer mixing ratios were at times much higher than they were in the spring in both 2014 (Mungall et al., 2016) and 2016 (unpublished results), reflecting nearby marine sources. For example, high atmospheric DMS concentrations (up to 1800 ppt, median 144 ppt) were found within the boundary layer from ship-based grab samples collected in July and August 2016. For a similar period and location in 2014, these values were up to 1100 ppt (median 186 ppt) (Mungall et al., 2016). Biogenic DMS oxidation products were also prevalent in the marine boundary layer (Ghahremaninezhad et al., 2016).

VOCs were measured in the marine atmosphere during the 2014 CCGS *Amundsen* cruise. Isoprene and monoterpene levels were frequently below detection limit, but occasionally reached as high as 15 ppt (Mungall et al., 2016). Oxygenated volatile organic compounds (OVOCs) were also measured, using an instrument that is especially sensitive to organic acids. High levels of formic acid (up to 4 ppb) and isocyanic acid (up to 80 ppt) were strongly correlated with a suite of C4–C11 oxo-acids (Mungall et al., 2017). Using positive matrix factorization, these OVOCs, which were elevated in regions where the ocean had high dissolved organic carbon (DOC) content, were interpreted as originating from an ocean source (Fig. 4). Production at the sea surface microlayer was invoked as an explanation, because compounds like formic acid are sufficiently soluble that they should not escape from the bulk ocean into the atmosphere. Rather, these species must arise either through photochemistry or heterogeneous oxidation proceeding in the sea surface microlayer, or by gas-phase atmospheric oxidation of components volatilized from the microlayer. Although the OVOC molecules measured by Mungall et al. (2017) were too small to participate in formation of Arctic marine secondary organic aerosol (MSOA) themselves, similar processes that form larger, less volatile molecules could contribute to aerosol growth. Formation of Arctic MSOA and its role in new-particle growth in the Arctic is discussed further in Section 4.2. [Note that in this work we use the term Arctic Marine Secondary Organic Aerosol to refer to the organic aerosol formed in the atmosphere from marine biogenic emissions. We note that the chemical character of Arctic MSOA is not necessarily the same as that in other marine environments. For example, different biogenic precursors may be present elsewhere, and the SOA that forms from shipping emissions will have very different properties and composition.]

NETCARE provided the opportunity to make some of the first observations of ammonia in the Arctic atmosphere. Previous measurements of atmospheric ammonia over the Norwegian Sea and Arctic Ocean during the summer ranged between the



detection limit (35 ppt) and 400 ppt (Johnson et al., 2008). Simultaneous measurements of sea surface ammonium ($NH_x$) during these previous studies ranged between 29 and 616 nM, leading to ammonia compensation points that were below the ambient concentrations, suggesting that the ocean could not act as a source of ammonia to the atmosphere. During the 2014 NETCARE campaign, hourly atmospheric ammonia measurements in the CAA marine boundary layer ranged between 40 and 870 ppt

(Wentworth et al., 2016). Simultaneous measurements of $NH_x$ at the sea surface and in melt ponds confirmed that these reservoirs could not act as sources of ammonia to the atmosphere. Boreal fires contributed to elevated atmospheric $NH_3$ in the CAA during 2014 (Lutsch et al., 2016), but could not fully explain the spatial and temporal extent of the elevated $NH_3$ mixing ratios. The inclusion of $NH_3$ emissions from migratory seabird colonies in model simulations brought predicted $NH_3$ values into much better agreement with observations (Wentworth et al., 2016) and strongly influenced modelled new particle

formation (Croft et al., 2016a). In 2016, observations were again made from the CCGS *Amundsen* but at a higher time resolution, as well as at the Alert field site, both from mid-June to mid-July (Murphy et al., 2018). The ranges of hourly average $NH_3$ values measured from the ship in 2016 (up to 1150 ppt; median 125 ppt) and at Alert (up to 720 ppt; median 234 ppt), were similar to the observations in 2014. Limited measurements of the tundra soil emission potential at the Alert site indicated that under the unusually high temperatures experienced at Alert in July 2016, the tundra could act as a source of ammonia to

the atmosphere. Overall, the bi-directional exchange of ammonia between the atmosphere and the land-ocean surface is important to include in chemical transport models. The impact of ammonia on aerosol formation in the summertime Arctic, with associated climate impacts, is discussed below in Section 4.2.

**3.4 Connecting the ocean, sea ice and the atmosphere through DMS modelling**

The un-extrapolated version of the Lana et al. (2011) DMS climatology averaged for the months of July and August clearly

demonstrates the scarcity of surface ocean DMS measurements in the Arctic during the most productive time of the year prior to the NETCARE field campaigns (Fig. 5a). Lack of data over the Canadian Polar Shelf and the Baffin Bay area challenged the representativeness of the standard (extrapolated) version of this climatology for these specific regions (Fig. 5c). Observations gathered through NETCARE field campaigns (Fig. 5b) significantly enhanced coverage in these regions. As part of NETCARE, a new process-based sea ice-ocean biogeochemical model representing ecosystems in both the sea ice

and water column of the marine Arctic was developed. The model was initially developed in a one-dimensional (1-D) configuration (Mortenson et al., 2017). Subsequently, sulfur and inorganic carbon cycling were developed and implemented into the model (Hayashida et al., 2017; Mortenson et al., 2018). The simulated Arctic sea ice ecosystem and sulfur cycle were next incorporated into a three-dimensional (3-D) regional configuration (Hayashida et al., 2018a, 2018b). This model advances previous Arctic-focused DMS model studies (Elliott et al., 2012; Jodwalis et al., 2000) in that many of the parameters

concerning the DMS production are derived from recent field observations in the Arctic, enabling quantification of the relative contributions of ice algae and phytoplankton to DMS production and emissions. The 1-D simulations indicated a notable contribution of ice algae: an 18 % enhancement of DMS concentrations under the ice and a 20–26 % enhancement of sea–air DMS fluxes during the melt period for Resolute Passage (Hayashida et al., 2017). Also in the vicinity of ice margins, simulated



spikes in sea–air fluxes of DMS originating from bottom and under-ice production by algae were comparable to some of the local maxima in the summertime flux estimated for ice-free waters in the Arctic.

The data obtained during the two NETCARE ship campaigns, together with data previously available in the PMEL sea surface database (https://saga.pmel.noaa.gov/dms/), were used to develop a new satellite-based model allowing the estimation of DMS

at the global and regional scales (Galí et al., 2018b). As can be seen in Fig. 5d, the satellite-based model provides a DMS map with an unprecedented spatial resolution (8 days, $28 \times 28$ km pixels). The DMS climatology based on the 3-D process-based model simulation shows a range similar to the Lana et al. (2011) climatology, but higher spatial variability, in line with the satellite-based climatology (Fig. 5e). Together with the remote sensing approach, the numerical model is being used to help interpret the new NETCARE DMS dataset, as well as to investigate longer-term and larger-scale variability, such as impacts

of sea ice reduction on DMS production (Hayashida et al., 2018b).

Under future global warming conditions, sea ice extent is expected to decline significantly, affecting the temporal and spatial evolution of ice algae and under-ice and open-water phytoplankton blooms. This may lead to changes in oceanic DMS emissions, although the sign and magnitude of the change is highly uncertain. Using the satellite approach mentioned above, Galí et al. (2018a) showed that DMS emission has increased at a rate of about 30 % per decade during the last two decades,

accompanied by large interannual changes linked to variable ice retreat patterns. They also estimated a 2- to 3-fold increase in DMS emission in response to complete sea ice loss in summer.

To estimate the sensitivity of Arctic aerosols and radiative forcing to surface seawater concentrations of DMS in the Arctic, simulations with different specified surface seawater DMS concentrations and spatial distributions in the Arctic were performed for future sea ice conditions using the Canadian Atmospheric Global Climate Model (CanAM4.3). For all of the

specified surface seawater DMS conditions in the model, simulated Arctic sulfate aerosol amounts respond only weakly to a reduction in sea ice extent owing to increases in precipitation and aerosol wet deposition associated with the receding ice and increased open water (Mahmood et al., 2018). However, nucleation rates for sulfate aerosol respond significantly to reductions in sea ice extent, which leads to a strengthening of cloud radiative forcing in the future. Furthermore, the simulated response of the mean cloud radiative forcing in the Arctic is proportional to the mean surface seawater DMS concentration in the Arctic.

Thus potential future changes in sea ice extent may result in a negative climate feedback of DMS on radiative forcing in the Arctic, as suggested by Charlson et al. (1987).

## 4 Arctic aerosol: sources, sinks, and properties

### 4.1 Rationale and research questions

The overall motivation of Arctic summertime research is to determine how the atmosphere will respond to melting sea ice, as

an ocean that was largely covered by sea ice through much of the summer will potentially be ice-free in summer by mid-century (AMAP, 2017; Comiso, 2011; Gregory et al., 2002; Holland et al., 2006). Given the evolution of the summertime Arctic Ocean from a bright ice cap to a dark ocean that can readily absorb solar radiation, it is of particular importance to



understand factors controlling the overhead aerosol and cloud that could mediate the positive radiative feedback of declining sea ice. Precipitation associated with low clouds and fogs is common in the summertime (Browse et al., 2014). Wet deposition is a highly efficient aerosol removal mechanism, giving rise to a clean boundary layer in which new particles may be formed or into which they may be input. In these clean boundary layers, increases in the numbers of particles acting as cloud

condensation nuclei (CCN) may increase longwave warming by clouds if the absolute concentrations of CCN are sufficiently low (Mauritsen et al., 2011); otherwise, increases in CCN concentrations lead to enhanced shortwave cooling. In this context, it is important to better understand the processes that give rise to new particle formation and growth to CCN sizes, and the associated impacts on clouds. For example, how do the emissions of biogenic gases described in Section 3 affect new particle formation and growth in such environments, and what is their importance relative to anthropogenic inputs from local shipping

and long-range transport?

In contrast, the springtime atmosphere, with its associated Arctic haze, has been better studied than the summertime atmosphere. The results from high profile campaigns such as ISDAC (Indirect and Semi-Direct Aerosol Campaign, acrf-campaign.arm.gov/isdac/), ARCTAS (Arctic Research of the Composition of the Troposphere from Aircraft and Satellites, www.nasa.gov/mission_pages/arctas/), and ARCPAC (Aerosol, Radiation, and Cloud Processes affecting Arctic Climate,

www.esrl.noaa.gov/csd/projects/arcpac/) have emphasized the importance of long-range transport (see also the POLARCAT project (Polar Study using Aircraft, Remote Sensing, Surface Measurements and Models, of Climate, Chemistry, Aerosols, and Transport, www.atmos-chem-phys.net/special_issue182.html)). However, many questions remain. Taking BC-containing aerosol as an example, we can ask: what is the relative importance of sources in Europe and different Asian regions, and how does the relative importance of different source regions vary vertically from the surface to higher altitudes? To what degree

can specific source regions be identified? How important are local sources, such as from Arctic shipping or gas flaring? How will the direct effect of light-absorbing particles be impacted by their mixing state, that is, by the degree to which they are internally or externally mixed with other components of the pollution plume? More generally, the composition of the air masses throughout the Arctic needs to be better evaluated vertically to aid in the identification of long-range transport sources, to help establish whether chemical transformations occur during transit and descent within the Arctic air mass, and to ultimately better

estimate climate impacts.

Lastly, the deposition rates of aerosol constituents need to be measured to better constrain models. Ideally, both wet and dry deposition rates would be individually evaluated throughout the year, to map out the transition from a system dominated by the relatively slow loss with ice cloud scavenging versus the more efficient removal via warm clouds and fogs.

## 4.2 Summertime aerosol: particle formation and growth

As described in Section 2, a pronounced Aitken mode in the aerosol size distribution is a common feature during the Arctic summertime, as demonstrated by Croft et al. (2016b), who identified this feature in long-term monitoring data sets from both the Alert and Zeppelin ground stations (Fig. 6). One of the major findings from NETCARE is the widespread prevalence of



5–50 nm ultrafine particles in the summertime Canadian Arctic (Burkart et al., 2017b; Collins et al., 2017; Willis et al., 2016, 2017) and their ability to activate as CCN (Burkart et al., 2017a; Chaubey et al., 2018). While previous ship-based measurements in similar regions in late summer and fall had demonstrated new particle formation and growth events, their frequency was low. For example, in the fall period of late August to the end of September 2008, only 3 such events were

observed over a five week observation period, whereas no events were observed at all in October 2007 (Chang et al., 2011b). By comparison, NETCARE measurements in mid-July to mid-August 2016 observed enhancements in 5–50 nm particles 41 % of the time in a spatially heterogeneous manner (Collins et al., 2017). Characterization of the summertime source of particles is provided in Fig. 7, wherein the number of particles between 15 and 30 nm (N15-N30) is highly enhanced at Alert in July and August, before rapidly declining in September (see Supplementary Information for details). NETCARE aircraft

measurements in July 2014 also demonstrated the spatial heterogeneity of 5–50 nm particle numbers in the inversion layer, with the highest concentrations observed over marine and cloudy regions and little detectable enhancement over ice-covered areas (Burkart et al., 2017b). These aircraft measurements also indicate that the numbers of these tiny particles in the free troposphere are spatially homogeneous and considerably lower than those measured in the inversion layer, indicative of a boundary layer source.

Significant growth of 5–50 nm particles to CCN sizes was clear from each observational platform. At Alert (Fig. 7), the summertime enhancement in particles between 15 nm and 30 nm (N15–N30) coincides with the increase of particles in the 50 to 100 nm size range (N50–N100), which is also the size of particle activation diameters observed in the field (see Section 4.3). Interestingly, using FTIR absorption of particulates collected on filters, the ratio of aerosol organic material to sulfate was also observed to increase during this time period, and the region of amide functional groups indicates a contribution of

organic components from breakdown of seabird urea in guano (Leaitch et al., 2018b). Likewise, a particle growth episode was clearly observed over the ice-free Lancaster Sound, in which the numbers of 5–50 nm particles and CCN increased in concert with the measured organic content of the PM1 aerosol (Willis et al., 2016). Across the entire aircraft campaign, the numbers of CCN were most strongly enhanced above background levels when the air had recently been at low altitude over open water (Fig. 8a), when the wind speeds were low, and when the organic-to-sulfate ratio of the particles was high (Willis et al., 2017).

This marine influence is consistent with summertime single-particle mass spectrometric measurements of trimethylamine-containing particles in the marine boundary layer that were largely externally mixed with sea-salt-containing particles (Fig. 9) (Köllner et al., 2017).

The lack of a wind speed dependence and the observations of externally-mixed particulate trimethylamine suggests that secondary sources are important. Similarly, microphysical models of growing particle size distributions could only be

reconciled with observations from the CCGS *Amundsen* icebreaker in northern Baffin Bay by invoking partitioning of semi-volatile species to the freshly nucleated and pre-existing particles (Burkart et al., 2017a). This stands in contrast to mid-latitude continental settings, where the growth behaviour is best modelled by considerable condensation of low volatility species such as sulfuric acid and highly oxygenated organic molecules. We presume this semi-volatile material is organic in nature (i.e., Arctic MSOA).



Natural emissions of ammonia are also important in this context. Wentworth et al. (2016) used GEOS-Chem model simulations to interpret NETCARE ammonia measurements (see Section 3.3) and found that migratory seabird colonies were important sources of ammonia in the summertime Arctic. In addition, transport of boreal wildfire smoke from lower latitudes can also be an important, albeit episodic, contributor of ammonia to the summertime Arctic troposphere (Croft et al., 2016a; Lutsch et

al., 2016; Wentworth et al., 2016). Croft et al. (2016a) further interpreted NETCARE observations using the GEOS-Chem-TOMAS model to find that ammonia from seabird-colony guano is a key factor contributing to the bursts of newly formed particles that are observed every summer at Alert (Fig. 10). In addition, the FTIR absorption in the region of amide functional groups indicates a contribution of organic components from the breakdown of seabird urea in guano. The chemical transport model simulations indicate that the pan-Arctic seabird-influenced particles can grow by sulfuric acid and organic vapour

condensation to diameters sufficiently large to promote pan-Arctic cloud droplet formation and effects on climate in the clean Arctic summertime. Other natural ammonia sources, including but not limited to biomass burning (Lutsch et al., 2016) and tundra emissions (Murphy et al., 2018), could also contribute to these effects (Croft et al., 2018).

Determining the precursors to Arctic MSOA is of crucial importance. Aerosol mass spectrometry measurements from the aircraft campaign in summer 2014 indicate that the organic chemical character of this aerosol is distinctly different from that

which arises from oxidation of common continental precursors, such as isoprene or the monoterpenes (Willis et al., 2017). The mass spectral signatures indicate molecules that instead have substantial alkyl components, such as functionalized fatty acids (Fig. 8b). Long-chain fatty acids can sometimes be a significant component of the sea surface microlayer (Cunliffe et al., 2013). Croft et al. (2018) have shown that a steady flux of condensable organic material from the ocean that oxidizes with a lifetime of a day is essential for consistency between GEOS-Chem-TOMAS modelled aerosol size distributions and those

measured at Alert and from the CCGS *Amundsen* icebreaker. This evidence strongly supports the importance of Arctic MSOA in setting the overall aerosol composition and size in the summertime.

## 4.3 Summertime aerosol: impacts on liquid water clouds

Studies at mid-latitudes have routinely shown that the smallest particles that can serve as nuclei for liquid cloud droplets are 80–120 nm diameter (Hoppel et al., 1985; Leaitch et al., 1986). The smaller Aitken particles, 20–80 nm in size, are commonly

considered to be too small to activate into cloud droplets. However, there are two circumstances which enable Aitken particles to activate at cloud base: 1) rapid cooling rates, generally associated with higher updraft speeds, that increase cloud base supersaturation; and 2) very low concentrations of larger particles (>100 nm), which reduce water vapour uptake at cloud base, thereby increasing supersaturations. In the second case, which is prominent in the Arctic during summer, modelling suggests that even modest updrafts (20–50 cm s$^{-1}$) lead to the activation as CCN of particles as small as 40 nm (Korhonen et al., 2008b,

2008a). This had never previously been verified by observations and was a main focus of the NETCARE summertime flight campaign.

During the NETCARE flights conducted out of Resolute Bay in July 2014, number size distributions of cloud droplets and aerosol particles measured in and around clouds showed that 50 nm particles were routinely activated and that particles as



small as 20–30 nm were activated on a few occasions where updraft speeds were likely higher (Leaitch et al., 2016). These results substantiate the prediction made by Korhonen et al. (2008b). However, Leaitch et al. (2016) found no evidence for an association of cloud liquid water content with aerosol variations when droplet concentrations are less than about 10 cm$^{-3}$, which was proposed by Mauritsen et al. (2011) as a means of aerosol-induced longwave warming. Modelling conducted as part of

NETCARE demonstrated the importance of this Aitken particle activation. For example, as mentioned above, Croft et al. (2016a) estimated the Arctic summertime shortwave radiative forcing by the effects of natural seabird ammonia emissions on these particles at −0.5 W m$^{-2}$, highlighting the importance of this natural aerosol for climate.

Lastly, experiments are in progress to evaluate the Single Column Model of Arctic Boundary Layer Clouds (SCM-ABLC) and version 18 of the Canadian Climate Centre's radiative transfer model with the cloud observations conducted from Resolute

Bay discussed above. The modelling work will attempt to reproduce the observations and quantify the uncertainty in modelling the radiative balance of low clouds and fog in the summertime Arctic.

### 4.4 Springtime aerosol: sources and vertical distribution

As discussed in Section 2, Arctic haze is a prominent feature in springtime, yet its composition and sources remain uncertain. During the NETCARE 2015 aircraft campaign, vertically resolved observations of trace gases and aerosol composition were

made in the high Arctic springtime, with six flights north of 80° N. Trace gas gradients observed on these flights defined the polar dome (i.e., the region north of the Arctic front) as north of 66–68.5° N and below potential temperatures of 283.5–287.5 K (Fig. 2) (Bozem et al., 2018; Willis et al., 2018).

NETCARE flight observations based at Alert and Eureka revealed that within the polar dome, submicron aerosol composition varied systematically with potential temperature. In the lower polar dome (i.e., below 252 K), measured aerosol mass (non-

refractory aerosol and BC) was dominated by sulfate (74 %), with smaller contributions from BC (1 %), organic aerosol (OA, 20 %) and ammonium (NH$_4$, 4 %). At higher altitudes and warmer potential temperatures, BC, OA and NH$_4$ contributed up to 3 %, 42 % and 8 % of aerosol mass, respectively. These observations indicate a substantial but unquantified contribution from sea salt aerosol in the lower polar dome (Leaitch et al., 2018b; Willis et al., 2018). Vertically resolved observations suggest that measurements at the surface may underestimate the contribution of OA, BC and NH$_4$ to aerosol transported to the Arctic

troposphere (Schulz et al., 2018; Willis et al., 2018). Next, we discuss hypotheses that may explain this vertical variability in aerosol composition.

Model simulations of air mass history using FLEXPART indicate differences in transport history as a function of potential temperature in the polar dome. The coldest and driest air masses, which also had the largest fraction of sulfate aerosol, spent long times (>10 days) in the polar dome, while warmer air masses showed some sensitivity to the surface at lower latitudes

(Willis et al., 2018). Model results indicate that descent of air masses from higher potential temperatures influenced the lower polar dome on the timescale of 10 days. Submicron aerosol composition varied systematically with model-predicted time spent in the mid-to-lower polar dome (i.e., below 265 K): the sulfate fraction increased with time below 265 K, while the NH$_4$, OA and BC fractions decreased significantly. These phenomena could arise from a combination of three possible processes: 1)



systematic changes in source region with increasing potential temperature (Fisher et al., 2011) that supply aerosol with systematically different compositions; 2) oxidation of transported aerosol and sulfur dioxide over the long aerosol lifetime in the polar dome and during transport; and 3) wet removal and cloud processing along the transport path that may impact the composition of aerosol arriving in the polar dome.

An analysis of results from simulations with four different models in NETCARE (Mahmood et al., 2016) indicates that the main source of BC in the Arctic is long-range transport from mid-latitudes. The long-range transport of BC to the Arctic is particularly efficient in midwinter and then decreases in efficiency, reaching a minimum in March and April. At the same time, dry deposition decreases, and wet deposition from clouds in the low and mid troposphere becomes more important during the transition from winter to spring. Overall, net changes in sources and sinks of BC in the Arctic are small, leading to nearly

steady Arctic burdens for this time period. Subsequently, during the transition from spring to summer, precipitation increases and wet deposition becomes highly efficient, which leads to substantial reductions in BC burdens in the Arctic despite increased emissions from vegetation fires. At high altitudes in the Arctic, the model results indicate that convective transport of pollution from the lower to the upper troposphere at lower latitudes and subsequent long-range transport to the Arctic represents an important source of BC.

Xu et al. (2017) interpreted a series of airborne and ground-based BC measurements made using multiple measurement techniques with the GEOS-Chem global chemical transport model and its adjoint to attribute the sources of Arctic BC (Fig. 11). This was the first comparison of BC measurements from a Single Particle Soot Photometer (SP2) at Alert with a chemical transport model. The inclusion of seasonally varying domestic heating and of gas-flaring emissions was crucial to successfully simulating ground-based measurements of BC concentrations at Alert and Barrow and airborne BC measurements across the

Arctic. Sensitivity simulations suggest that anthropogenic emissions in eastern and southern Asia have the largest effect on the Arctic BC column burden in spring (56 %), with the largest contribution in the middle troposphere. At the Arctic surface, anthropogenic emissions from northern Asia (40–45 %) and eastern and southern Asia (20–40 %) are the largest BC contributors in winter and spring, followed by Europe (16–36 %). This modelled source distribution in Asia and Europe is different from that found in previous studies (Bourgeois and Bey, 2011). The adjoint simulations indicate pronounced spatial

heterogeneity in the contribution of emissions to the Arctic BC column concentrations, with noteworthy contributions from emissions in eastern China (15 %) and western Siberia (6.5 %). Emissions from as far away as the Indo-Gangetic Plain could have a substantial influence on Arctic BC as well.

## 4.5 Springtime aerosol: optical properties

Kodros et al. (2018) combined measurements of BC mixing state in the springtime Canadian high Arctic with simulated size-

resolved aerosol mass and number concentrations to constrain model estimates of the direct radiative effect (DRE). Airborne measurements using an SP2 and Ultra-High Sensitivity Aerosol Spectrometer onboard the Polar 6 aircraft show an average coating thickness of 45 to 40 nm for BC core diameters across the range of 140 to 220 nm, respectively. For total particle diameters ranging from 175 to 730 nm, BC-containing particle number fractions range from 16 % to 3 %. GEOS-Chem-



TOMAS yields a pan-Arctic average springtime DRE ranging from −1.65 W m$^{-2}$ for entirely externally mixed BC to −1.34 W m$^{-2}$ for entirely internally mixed BC. Using the observed mixing-state constraints from this field campaign significantly reduces this estimated range in DRE by over a factor of two (−1.59 to −1.45 W m$^{-2}$). Measurements of mixing state thus provide important constraints for model estimates of the DRE.

5 Some of the first vertically-resolved and concurrent measurements of aerosol composition and optical properties in the springtime high Arctic are presented in Leaitch et al. (2018a). As shown in Fig. 12a, observations from the POLAR 6 during April 2015 indicate an increase in the fraction of rBC in submicron particles with altitude coincident with an increase in the overall carbonaceous fraction of the submicron aerosol for flights conducted around Alert, Nunavut and Eureka, Nunavut (Schulz et al., 2018; Willis et al., 2018). For values of the light scattering coefficient ($\sigma_{scat}$) less than 15 Mm$^{-1}$, which represent 98 % of the measured $\sigma_{scat}$ during the Alert and Eureka flights, the single scattering albedo (SSA) of the aerosol decreases from 0.96 near the surface to 0.93 at 500 hPa (Fig. 12a). The SSA decrease with altitude is consistent with the increasing fraction of rBC in the particles and suggests a stabilizing influence of BC on the high Arctic atmosphere. In an absolute sense, the $\sigma_{scat}$ values primarily vary with the sum of ammonium, organics and sulfate as shown in Fig. 12b.

**4.6 Monitoring the transitions between seasons by remote sensing**

15 While in situ field campaigns provide detailed information over a short period of time, remote sensing provides annual measurements and thus information about the transitions from winter to spring and then into summer. In particular, ground-based lidar and starphotometry (carried out at the PEARL observatory in Eureka, Nunavut) and satellite-based lidar (CALIOP/CALIPSO) during the latter half of two polar winters suggest the frequent Arctic-wide presence of submicron particles in the boundary layer with aerosol optical depths (AOD) significantly greater than the AOD predicted by GEOS-Chem, in which the AOD largely results from sulfate particles (O'Neill et al., 2016). Ground-based sunphotometry (AEROCAN/AERONET) measurements acquired between 2009 and 2012 at five western-Arctic stations (Hesaraki et al., 2017) revealed Arctic-wide springtime peaking of both submicron and supermicron AODs that were roughly consistent with submicron and supermicron AOD estimates from GEOS-Chem (predominantly associated with Arctic haze sulfates and Asian mineral-dust aerosols, respectively). A summertime peak in submicron particles, which was determined to be smoke-induced at the four western-most AEROCAN Arctic stations, was not simulated by GEOS-Chem. Rather, GEOS-Chem indicated a continuous spring-to-fall decrease in submicron AOD that was principally associated with a decrease in sulfate contributions.

**4.7 Aerosol deposition to snow**

Deposition fluxes in the Arctic are very poorly characterized, in large part because of the logistical challenges of collecting continuous data series. To address this lack, NETCARE scientists collected temporally resolved data for the chemical composition of snow (common metals, BC, soluble ions, and small organics) that fell throughout the cold season at Alert



(Macdonald et al., 2017). In particular, new snow was collected after each appreciable period of precipitation, resulting in samples every four days, on average, from September 2014 to May 2015.

Using measurements of the amount of snow that had fallen in a given area, the chemical compositions were converted to fluxes for comparison with modelled values. In combination with ambient air concentrations of the equivalent chemical species, the

measured fluxes were then expressed as an effective deposition velocity, which encompasses both wet and dry deposition processes (Fig. 13) (Macdonald et al., 2017). Interestingly, effective deposition velocities are higher for the warmest months (Sept, Oct, May) than for the cold months, arising from the switchover from liquid water to a combination of dry deposition and ice cloud scavenging. The effective deposition velocities for BC were the smallest of all species characterized, consistent with its low hygroscopicity and poor ice-nucleating abilities.

To take advantage of the high temporal resolution of the samples, the data were also used to assess potential sources contributing to chemical species in snow using a combination of positive matrix factorization and FLEXPART potential emission sensitivity analysis (Macdonald et al., 2018). The best positive matrix factorization solution consisted of seven source factors (sea salt, crustal metals, BC, carboxylic acids, nitrate, non-crustal metals, and sulfate), reflecting a balance between natural and anthropogenic sources. Notable findings include identification of anthropogenic sources (but not biomass burning)

as dominant for BC during this study period, and a potential source of volcanic sulfur in the fall of 2014.

A simple parameterization of BC in snow was developed and tested in the Canadian Atmospheric Global Climate Model (CanAM). According to the parameterization, the temporal evolution of the concentration of BC near the top of the snowpack is determined by changes in dry and wet deposition of BC, the snowfall flux and scavenging by snow meltwater. Comparison of model results with a multi-year climatology of BC concentrations in snow produces good agreement for locations in the

Canadian Arctic and sub-Arctic (Doherty et al., 2010, 2014; Wang et al., 2013b) as well as for other regions in the Northern Hemisphere (Namazi et al., 2015). Simulated changes in BC loading in snow in the second half of the $20^{th}$ century had much smaller cryospheric impacts on surface air temperatures than other aerosol and greenhouse gas radiative forcings.

## 4.8 Ship emissions

Understanding the impacts of ship emissions on climate and air quality of the Arctic environment is challenging but important,

given the likelihood of future increases in Arctic shipping. The Arctic atmospheric boundary layer exhibits different dynamics from mid-latitudes, being characterized by thermally stable conditions with reduced turbulent mixing (Aliabadi et al., 2016a). Ships navigating northern latitudes operate under partial engine load and icebreaking conditions as opposed to full speed cruising. Uncertainties are compounded by the lack of accurate predictions for increased ship traffic patterns in the Arctic as the ice cover retreats, as well as the lack of a robust regulatory framework to control emissions via Emission Control Areas set

by the International Maritime Organization (Aliabadi et al., 2015).

The NETCARE campaign near Resolute Bay in July 2014 characterized typical ship emissions and plume evolution by mapping the plume of the CCGS *Amundsen* with the Polar 6 research aircraft (Aliabadi et al., 2016b). Three plumes were intercepted, with the first corresponding to operation of the CCGS *Amundsen* in open water under low-speed cruise conditions,



while the second and third corresponded to operation under icebreaking conditions. The measured species included mixing ratios of $CO_2$, $NO_x$, CO, $SO_2$, particle number concentration, BC, and CCN. Lower plume expansion rates were observed compared to mid-latitudes due to reduced turbulent mixing, resulting in a poorly diluted plume that was confined within a low boundary layer. Most, but not all, emission factors agreed with prior observations for low engine loads at mid-latitudes. This

implied different emission factors for each species measured. In particular, icebreaking increased the $NO_x$ emission factor to values equivalent to those measured for high engine loads at mid-latitudes, likely due to differences in engine combustion temperatures. The CO emission factor was higher at low engine loads, whereas the BC emission factors were similar to those at mid-latitudes; the effect of engine load on BC emission factors is still debated in the literature. Due to the use of low sulfur diesel fuel by the CCGS *Amundsen*, no $SO_2$ was detected.

## 10   5 Ice nucleating particles

### 5.1 Rationale and research questions

Currently, clouds are responsible for some of the greatest uncertainties in climate change predictions. This is in large part because the properties of clouds and their formation processes are poorly understood, especially in the case of ice and mixed-phase clouds (Cantrell and Heymsfield, 2005; Hegg and Baker, 2009; Murray et al., 2012). Particles that can initiate ice

formation in the atmosphere at temperatures and relative humidities lower than those required for homogeneous freezing of solution droplets are referred to as ice nucleating particles (INPs) (Vali et al., 2015). Only a very small fraction of atmospheric particles (1 in $10^{-3}$ to $10^{-5}$) can act as INPs (Rogers et al., 1998), but predictions of climate and precipitation can depend strongly on INP concentrations (DeMott et al., 2010; Lohmann, 2002). As an example, an increase in the concentrations of INPs can lead to more precipitation and shorter cloud lifetimes for mixed-phase clouds, resulting in less solar reflectivity (DeMott et al.,

2010; Lohmann, 2002). Despite the importance of INPs, the level of scientific understanding of their concentrations and sources in the atmosphere remains low (Coluzza et al., 2017). To improve predictions of precipitation and climate in the Arctic, the concentrations and sources of INPs in this region need to be determined. This information can then be used to test and constrain parameterizations in atmospheric models (Vergara-Temprado et al., 2017).

### 5.2 INPs in the sea surface microlayer and bulk sea water

The sea surface microlayer is the interface between the atmosphere and the ocean and is a source of particles to the atmosphere via wave-breaking and bubble-bursting. INPs have previously been detected in bulk seawater (Schnell, 1977; Schnell and Vali, 1975, 1976); however, concentrations and properties of INPs in the microlayer had not been confirmed prior to the start of NETCARE. This lack of information led to large uncertainties in quantifying the importance of the microlayer as a source of INPs to the atmosphere (Burrows et al., 2013). In initial experiments, the concentration of INPs in the microlayer collected off

the west coast of Canada were measured (Wilson et al., 2015), while in parallel, researchers from the University of Leeds measured the properties and concentrations of INPs in the microlayer collected off the east coast of the United States and



Greenland (Wilson et al., 2015). We built on this work by measuring the concentrations and properties of INPs in the microlayer collected in the Canadian Arctic (Irish et al., 2017, 2018b).

Microlayer samples were collected in the Canadian Arctic during the summers of 2014 and 2016 from the CCGS *Amundsen*. INPs were ubiquitous in the microlayer with freezing temperatures as warm as −5°C. Concentrations of INPs were higher on

average in 2016 than in 2014 or off the east coast of the US and Greenland (Wilson et al., 2015). The INP concentrations were enhanced in the microlayer compared to bulk seawater in several samples collected in 2016. Concentrations of INPs were anti-correlated with salinity, possibly indicating that the INPs were associated with melting sea ice. The INPs had diameters between 0.2 and 0.02 µm and were heat-labile, and therefore likely biological material. Possible candidates for the INPs include exudates from sea-ice microorganisms such as sea-ice diatoms and bacteria.

**5.3 INPs in the high Arctic during spring–summer**

The size of INPs collected from the atmosphere at Alert in late spring and early summer 2014 were also measured (Mason et al., 2016). The size of atmospheric INPs can help distinguish which types of atmospheric particles are important as INPs. During this campaign, the average concentrations of atmospheric INPs were 0.05 L$^{-1}$, 0.22 L$^{-1}$, and 0.99 L$^{-1}$ at freezing temperatures of −15°C, −20°C, and −25°C, respectively. The median diameters of the INPs were 3.2 µm, 2.2 µm, and 0.83

µm at freezing temperatures of −15°C, −20°C, and −25°C, respectively, and the average fractions of INPs ≥ 1 µm were 95 %, 66 %, and 41 % at freezing temperatures of −15°C, −20°C, and −25°C, respectively. These results suggest that the major sources of the INPs at this site during the collection period were not submicron aerosol particles, such as ammonium sulfate and BC particles. The sizes of the INPs measured at Alert were consistent with those of INPs measured at five other sites in North America, as well as one in Europe (Mason et al., 2016).

During March 2016, INP measurements at Alert were made daily (Si et al., 2018). In these high frequency data, INP concentrations were strongly correlated with tracers of mineral dust, suggesting that it was a major source of the INPs measured. These results are consistent with the size of INPs measured at Alert during the spring and summer of 2014. Particle dispersion modelling suggests that the mineral dust may have been transported over long distances from the Gobi Desert.

**5.4 INPs in the summertime marine boundary layer in the Canadian Arctic Archipelago**

During the summer of 2014, we measured atmospheric concentrations of INPs in the Canadian Arctic marine boundary layer on board the CCGS *Amundsen* (Irish et al., 2018b). Concentrations averaged 0.005 L$^{-1}$, 0.044 L$^{-1}$, and 0.154 L$^{-1}$, at freezing temperatures of −15°C, −20°C and −25°C, respectively. These values fell within the range of atmospheric concentrations measured at locations outside the Arctic and in the marine boundary layer (DeMott et al., 2016). Based on a combination of surface area measurements of mineral dust and sea spray aerosol (Fig. 14) and particle dispersion modelling using FLEXPART,

mineral dust from local sources is a more important contributor than sea spray aerosol to the atmospheric INP population for the times and locations studied. These results do not rule out the sea surface microlayer as a source of INPs to the Arctic marine




boundary layer; rather, they show that the sea surface microlayer is likely a smaller source of atmospheric INPs compared to local mineral dust for the locations and times studied.

## 5.5 Measurements of thin ice cloud microphysics linked to INP properties

Sulfuric acid coatings strongly affect INPs and thus their effect on clouds and precipitation. This is particularly important during Arctic haze events. Laboratory studies (Eastwood et al., 2009), *in situ* measurements (Jouan et al., 2014) and large-scale observations from the CloudSat and CALIPSO satellites (Grenier et al., 2009; Grenier and Blanchet, 2010) support the hypothesis of a dehydration-greenhouse feedback (Blanchet and Girard, 1994) linking acidified aerosols to the favoured formation of larger ice crystals and light precipitation through a reduction of INP activity. In cold Arctic conditions, thin ice clouds (TIC), like cirrus, are ubiquitous in the coldest free troposphere (Grenier et al., 2009). Two types have been identified: TIC-1, which has many small crystals (smaller than ~30 µm) and TIC-2, which has fewer but larger ice crystals. While TIC-1 is largely non-precipitating, TIC-2 leads to light precipitation, often in the form of diamond dust, which is sometimes called clear sky precipitation because of the very low optical depth of the clouds. Acidic INPs favour the formation of TIC-2 in the mid and upper troposphere, which enhances water flux towards the lower layers and leads to dehydration of the upper cold troposphere. In turn, this process lowers the greenhouse effect of water vapor and favours the direct IR cooling of the air mass in the lower atmosphere and at the surface. Hence, variations in the INP composition can significantly affect the radiative properties of the polar atmosphere and clouds, as well as the atmospheric moisture concentration.

A far IR radiometer (FIRR) developed with Canadian Space Agency support and especially designed to measure TIC properties and water vapor was flown on board the Polar 6 aircraft during the NETCARE campaign of April 2015. The goal was to simultaneously measure, for the first time, INPs, cloud microphysics and spectrally resolved radiation in the range of 8 to 50 µm over the Arctic. The experiment successfully demonstrated closure between measurements and theoretical calculations for clear sky conditions (Libois et al., 2016a, 2016b). It also showed the strong sensitivity of FIRR observations to ice crystal size and cloud optical depth. However, the limited number of ice clouds encountered and the complexity inherent in probing them with an aircraft highlighted the need for further campaigns dedicated to simultaneous investigation of ice cloud radiative and microphysical properties (Libois et al., 2016b). The results obtained from the NETCARE campaign have paved the way for a future satellite-based deployment over the poles, linking cloud microphysics, the atmospheric water budget, and radiation balance.

## 6 Remaining uncertainties in Arctic aerosol research

The NETCARE research outlined above has provided novel insights into 1) the biogenic sources of gases that can impact the size and composition of Arctic aerosol; 2) new particle formation and growth into sizes that were demonstrated to be activating to form cloud droplets, with growth occurring largely via formation of Arctic marine secondary aerosol; 3) the sources and properties of Arctic haze aerosol, in particular BC-containing particles; 4) Arctic INPs in the air, ocean water, and the sea




surface microlayer; and 5) deposition rates of pollutants to snow. Many of these advances arose as a result of the highly interdisciplinary approach taken within NETCARE. Nevertheless, despite these advances, many research questions remain, as outlined in this section.

## 6.1 Marine and coastal biogenic aerosol precursors

Observations gathered during NETCARE provide a valuable benchmark upon which to base predictions of the changes in the source strength of secondary MBA precursors in response to alterations in Arctic climate. This is important to determining the amounts of both aerosol sulfate and organics. The thinning and loss of seasonal sea ice, which is driven by warming and polar amplification, is by far the most conspicuous of these alterations (AMAP, 2013; Comiso, 2011; Serreze and Barry, 2011). The marine production of DMS could be particularly sensitive to both modifications in seasonal ice extent and the intermittent

presence of melt ponds above the ice in spring and summer (Gabric et al., 2017; Gourdal et al., 2018; Lizotte et al., 2018). Modelling and observational studies suggest that the northward shrinking of the seasonal ice extent and the ensuing increase in open waters available for gas exchange could lead to heightened primary productivity (Arrigo and van Dijken, 2015; Gabric et al., 2017; Ito and Kawamiya, 2010) and production of DMS (Levasseur, 2013). In turn, this would lead to higher atmospheric MSA and secondary sulfate (Sharma et al., 2012), and background particle concentrations (Dall´Osto et al., 2017). Simulations

with CanAM indicate that associated increases in concentrations of CCN could potentially offset part of the warming due to enhanced cloud albedo (Mahmood et al., 2018). Specifically, the projected loss of sea ice between 2000 and 2050 leads to a substantial increase in Arctic DMS emissions in CanAM, leading the cloud radiative forcing associated with Arctic DMS to increase from $-0.13$ W m$^{-2}$ to $-0.27$ W m$^{-2}$ during this time period if marine production of DMS is unchanged. Adding to these wide-ranging observations and modelling outputs, NETCARE results suggest that as seasonal (i.e., first-year) sea ice becomes

a pan-Arctic feature in the future (AMAP, 2017), ice-related sources of DMS could increase in response to the thinning of sea ice, as well as to the areal and temporal expansion of melt ponds that act both as a source of DMS and as a catalyst of under-ice bloom development. Conversely, observational and modelling studies also suggest that increased stratification in ice-free waters of the Arctic could curb primary productivity due to nutrient limitation (Steiner et al., 2015) and that wind-induced sea spray may be more prevalent in open waters, acting as a condensation sink for material that could form secondary aerosol

(Browse et al., 2014)

The Arctic system may also be vulnerable to other changes, notably ocean acidification, as well as amplified warming and freshwater inputs (AMAP, 2013; Yamamoto-Kawai et al., 2009). An experimental assessment of the impact of ocean acidification on DMS-producing planktonic communities of Baffin Bay during NETCARE (Hussherr et al., 2017) revealed that DMS production may decrease by 25 % under end-of-the-century scenario reductions of pH ($\Delta$pH$_T$ = $-0.48$), confirming

results observed in another Arctic study in the Svalbard Archipelago which showed a 35 % decrease in DMS production (Archer et al., 2013). Other studies, however, have suggested that organisms thriving in Arctic waters may already be resilient to moderate or acute natural fluctuations in pH, exhibiting high capacity to compensate for modifications in pH (Hoppe et al.,



2018) and no significant changes in DMS following acidification (Hopkins et al., 2018). Further experimentation is needed to identify the underlying causes for these contrasting DMS responses to ocean acidification in Arctic waters.

NETCARE illustrated for the first time the influence of ammonia emissions from seabird colonies on not only atmospheric mixing ratios (Wentworth et al., 2016) but also new particle formation, aerosol neutralization and associated indirect effects
on climate (Croft et al., 2016a, 2018). How will these emissions evolve with climate change and potential changes in wildlife populations (Weimerskirch et al., 2018), habitat, and migratory patterns? NETCARE measurements from Alert suggest that Arctic soils may also be an ammonia source (Murphy et al., 2018), perhaps reflecting the redistribution of ammonia between different components of the Arctic land-ocean ecosystem. This highlights the importance of including bi-directional fluxes in chemical transport models for species that move readily between the land, atmosphere, and ocean.

**6.2 Particle and SOA formation in summertime Arctic marine environments**

Unlike lower-latitude marine boundary layers (Quinn and Bates, 2011) particle nucleation and growth was frequently observed during NETCARE campaigns in the boundary layer in Arctic marine and coastal regions. The Arctic may behave differently for a number of reasons: 1) the persistent temperature inversion lowers the rate of mixing of surface emissions; 2) the summertime atmosphere has 24-hour sunshine to drive photo-oxidation; 3) the condensation sink is particularly low; and 4)
the low temperatures facilitate molecular cluster formation. It is crucial to assess the chemical components and particle formation mechanisms that prevail in this distinctive environment. Particularly valuable will be on-line mass spectrometric measurements of the chemical composition of the smallest clusters and particles that form at the early stages of the nucleation and growth process.

The growth of 5–50 nm particles into CCN size ranges was evident in multiple NETCARE campaigns (Burkart et al., 2017a,
2017b; Collins et al., 2017; Willis et al., 2016, 2017). Surprisingly, much of the sub-micron aerosol mass associated with growth was organic in composition (Burkart et al., 2017b; Willis et al., 2016, 2017), providing additional support to the idea that secondary organic aerosol of marine origin is important (Rinaldi et al., 2010). Although we do not know the precise nature of the organic precursors, the NETCARE studies described in Section 4.2 illustrated that the secondary organic source is marine and potentially associated with oxidation or photochemistry of the sea surface microlayer (see Sections 3.3 and 6.3). It
is important to determine the balance between secondary aerosol formation versus primary particle formation from sea spray. In one NETCARE case study of new particle formation and growth over Lancaster Sound, there were indications of secondary processes occurring alongside formation of sea spray salt particles, suggesting that these processes may sometimes occur simultaneously, complicating analyses (Collins et al., 2017; Köllner et al., 2017; Willis et al., 2016). Key uncertainties in the radiative effects of the Arctic MSOA simulation in GEOS-Chem-TOMAS include Arctic nucleation processes, the chemical
composition of Aitken particles, and the volatility of the SOA (Croft et al., 2018). Further understanding of these processes would better constrain climate feedbacks. We note that the composition and properties of Arctic MSOA are not necessarily the same as that formed in marine environments in other parts of the world.





### 6.3 The sea surface microlayer

Our understanding of how the sea surface microlayer impacts air–sea exchange of aerosols and gases is still largely circumstantial and is based mainly on conceptual models (Garbe et al., 2014; Lewis and Schwartz, 2013), laboratory experiments (Bigg and Leck, 2008; Wilson et al., 2015), and observations of similarities between particulate matter in the microlayer and the atmosphere (Leck and Bigg, 2005). Obtaining information on how these concepts play out in the real world has proven extremely challenging. That being said, one pronounced example of the potential importance of the sea surface microlayer comes from work in NETCARE that demonstrated that oxygenated VOCs in a marine Arctic setting were likely formed photochemically within the microlayer or by oxidation of gases arising from it (Mungall et al., 2017). Although laboratory studies have previously demonstrated that oxygenated VOCs can be chemically generated from microlayer surrogate materials (Rossignol et al., 2016; Zhou et al., 2014), such studies do not address the chemical complexity of the genuine environmental system.

Although additional experiments have previously quantified the impact of microlayer surfactant enrichment on gas exchange (Brockmann et al., 1982; Frew et al., 2004; Pereira et al., 2016), to date no one has directly tied natural variations in the sea surface microlayer to the exchange of aerosols or gases. The main difficulty lies in the different temporal and spatial scales of atmospheric and microlayer measurements. The composition of the microlayer is highly heterogeneous even on small horizontal scales (Cunliffe et al., 2013), and recovery of microlayer samples for chemical analysis is time consuming. Thus, tying those measurements to observations of temporally variable aerosols measured from ships or airplanes is innately subject to substantial uncertainties.

In order to confidently identify the relationship between the composition of the sea surface microlayer and atmospheric aerosol production, it will be necessary to collect coherent data sets from single platforms, such as autonomous surface craft (Ribas-Ribas et al., 2017). In addition, intelligently designed time series stations could provide sufficient data to identify clear relationships between the microlayer and the atmosphere (Cunliffe et al., 2013; Engel et al., 2017).

### 6.4 Removal of aerosol particles in the summertime

The 2014 NETCARE aircraft campaign illustrated that the low CCN numbers prevalent in the summer boundary layer can lead to large cloud droplet diameters, in some cases approaching 30 microns (Leaitch et al., 2016). The settling velocity of such droplets is sufficiently fast that drizzling low-level clouds and fogs play an important role in deposition to the surface (Browse et al., 2014). As yet, there is no Arctic deposition network that is quantitatively assessing the importance and efficiency of such processes. An important question that arises is the degree to which trends in wet scavenging are driving the trends in aerosol loadings. It is well documented that both aerosol sulfate and BC are currently lower in abundance than they were in previous decades (AMAP, 2015). To what degree is this trend due to reductions in source emissions as opposed to changes in cloud scavenging? In the summer in particular, the wider expanses of open ocean associated with sea ice melting may lead to higher water fluxes from the ocean to the atmosphere, potentially affecting cloud liquid water content and



deposition rates. Model simulations suggest enhanced wet deposition of sulfate aerosol by precipitation in reduced sea ice conditions (Mahmood et al., 2018). It is not known whether this enhanced deposition will affect sea ice melt rates.

## 6.5 Cloud scavenging and long-range transport

As described in Section 2, there is a transition in scavenging regimes between the efficient processes that occur with liquid clouds and the comparatively inefficient processes associated with pure ice clouds. However, the community struggles to accurately include such scavenging processes in models (Mahmood et al., 2016). This is important for transport of long-range pollutants from more industrialized southerly locations, and for the input of biomass burning aerosol that is likely to become more prevalent with the warming climate (Marelle et al., 2015; Shindell et al., 2008; Wang et al., 2013a).

The degree of aerosol scavenging that occurs outside the Arctic relative to that which occurs within it must be better established. For example, transport associated with warm conveyor belt systems is one mechanism that supplies pollutants to the Arctic (Ancellet et al., 2014; Roiger et al., 2011), while cloud formation associated with synoptic uplift in mid-latitudes cleans the air. This has been nicely demonstrated by a close inverse relationship between the accumulated precipitation along back trajectories, a measure of wet scavenging, and BC levels arriving in the Arctic (Matsui et al., 2011). Deciphering the efficiency of such extra-Arctic processes is one focus of the proposed IMPAACT project (https://pacesproject.org/abstract/introducing-impaact-investigating-pollutant-transport-asia-arctic-and-north-america), which will involve multiple research aircraft and surface-level vessels trying to better understand pollutant levels at their sources and their modifications along these transport pathways. A second example is wet scavenging that occurs as a result of air lifting over Arctic terrain, such as Ellesmere Island and Greenland. In ongoing NETCARE analysis, there is evidence that new particles formed in very clean free tropospheric air masses that had been lifted over Greenland and passed to its north. It is likely that cloud scavenging occurred over Greenland.

A related question is the degree to which oxidation processes modify the overall aerosol composition as a function of residence time in the Arctic. For the measurements described in Section 4.4, the increase of the sulfate-to-organic ratio with decreasing altitude in springtime aerosol may in part arise from formation of sulfate as the air mass ages. Validation of this mechanism awaits better $SO_2$ measurements.

## 6.6 INPs in the cold seasons and atmospheric impacts

While aerosol particle removal is exceedingly efficient under summer conditions, the ice nucleation processes in the colder months are much more selective and less well understood, as described in Section 5. We sought to understand which aerosol types contain the best ice-nucleating particles. Initial indications from NETCARE measurements are that dust is an important contributor to the INP population (Fig. 14), but that does not rule out the role of primary sea spray particles acting as INPs. A second important question was to what degree coatings of secondary materials, such as sulfates or organics, modify the ice nucleation properties of primary INPs, such as mineral dust. A major challenge is the development of better parameterizations of INPs for use in atmospheric modelling. To that end, work in NETCARE improved ice nucleation parameterizations in the




Global Multiscale Environmental Model (GEM-LAM) to determine the effect of pollution on clouds in the Arctic (Keita and Girard, 2016). To simulate pristine clouds, a parameterization of ice nucleation by mineral dust was included, whereas to simulate ice clouds influenced by pollution, a parameterization of ice nucleation by mineral dust coated with sulfuric acid was used. A parameterization was developed as well to test against the 2014 and 2016 CCGS *Amundsen* INP measurement data.

Nevertheless, many details about the ice cloud formation process are still missing from these parameterizations.

Although NETCARE measurements of aerosol deposition fluxes to the snow at Alert were made across a full cold season (see Section 4.7), the degree to which this flux occurred via dry or wet deposition was not precisely determined. In particular, it remains to be determined how important ice cloud scavenging and settling is as a particle removal process. A long-term, high time resolution aerosol deposition network that separates wet and dry deposition across the Arctic would be highly beneficial

in this regard.

## 6.7 Aerosol particle mixing state

Mixing state refers to the uniformity of the distribution of the aerosol chemical components across an array of particles; i.e., are all the particles of the same chemical composition or is their chemical distribution heterogeneous? NETCARE measurements have highlighted how this information is crucial to our understanding of aerosol sources and impacts. In

particular, the springtime measurements of BC aerosol described in Sections 4.4 and 4.5 showed that within the Arctic haze sampled in spring 2015, only 3–16 % of the particles contained BC inclusions and that BC-containing particles had coatings 40–45 nm in thickness on average (Kodros et al., 2018). The direct radiative forcing that is modelled using these results as constraints is distinctly different from that where it was assumed that the chemical mixing state of the aerosol was uniform (see Section 4.5). Likewise, in the summertime measurements from 2014, the single-particle mass spectrometry measurements

at low altitudes over open water illustrate that primary and secondary marine aerosol components were externally mixed, thus indicating different formation processes (Fig. 9) (Köllner et al., 2017; Willis et al., 2016). More measurements of this type, down to as small particle sizes as possible, are crucial to further determining the balance between primary and secondary marine aerosols, to establishing the degrees of coating that exist on mineral dust aerosol that contain INPs, and to assessing the efficiency of cloud scavenging that occurs across different particle types. For example, does the relatively hydrophobic

character of BC inhibit the rate at which it is wet cloud scavenged, and if so, how much hygroscopic coating material must be present to make the particles CCN active?

## 6.8 Measurements across the seasons and throughout the atmosphere

The Arctic springtime has been much more extensively studied than other seasons. This is understandable given the importance of the Arctic haze phenomenon. However, the fall and winter seasons are poorly characterized using intensive campaign

approaches, largely because of the operational difficulties in working under cold, dark conditions. Although remote sensing can be used to study transitions between seasons (see Section 4.6), it is still important to better understand how transport patterns of pollutants and their deposition rates change seasonally. As well, we know very little about the polar night and the





associated formation of ice clouds. The radiative effects of these clouds and their ability to dehydrate the atmosphere on a large scale through extensive light precipitation are important to assess. An exciting movement in this direction is the development of a far infrared radiometer (FIRR) that was flown on the Polar 6 aircraft for the first time within NETCARE (Libois et al., 2016a, 2016b) (see Section 5.5). By providing improved ice cloud characterization and measurements of

atmospheric water vapor, this instrument can be used to improve understanding of the cooling of the atmosphere via infrared emissions in cold polar regions.

The vertical profiles measured as part of NETCARE in both the springtime and summertime provide essential information for comparison to model outputs and provide a necessary complement to the much more extensive sets of measurements from ground-based field campaigns and stations. Additional aircraft campaigns that provide such vertically resolved features are a

top priority for future studies. Nevertheless, the continuous measurements at Arctic ground stations remain our most valuable data set to assess long-term trends. There is a significant need to enhance the instrumental capabilities at these stations, for example with key continuous measurements of $SO_2$, $NH_3$, VOCs and aerosol composition across all size ranges, to further our understanding of many of the issues described above.

**Author contributions**

JA is the NETCARE principal investigator. He coordinated and wrote substantial portions of the paper. All other co-authors contributed text and/or reviewed the paper. All co-authors either wrote a first-author paper as part of the NETCARE project or else contributed in a substantive manner to the research conducted in the project or presented in the paper. EG contributed to the INP and ice cloud research of NETCARE but he died before submission. We regard his approval for inclusion of his name on this paper as implicit.

**Acknowledgements**

NETCARE was funded by the Natural Sciences and Engineering Research Council (NSERC) of Canada under its Climate Change and Atmospheric Research Program, with additional financial and in-kind support from Environment and Climate Change Canada, Fisheries and Oceans Canada, the Alfred Wegener Institute, and the Major Research Project Management Fund at the University of Toronto. Colorado State University researchers were supported by the US Department of Energy's

Atmospheric System Research, an Office of Science, Office of Biological and Environmental Research program, under Grant No. DE-SC0011780, the U.S. National Science Foundation, Atmospheric Chemistry program, under Grant No. AGS-1559607, and by the U.S National Oceanic and Atmospheric Administration, an Office of Science, Office of Atmospheric Chemistry, Carbon Cycle, and Climate Program, under the cooperative agreement award #NA17OAR430001. All authors would like to greatly thank the editors for the NETCARE special issue in *Atmospheric Chemistry and Physics, Biogeosciences,* and



*Atmospheric Measurement Techniques* for their time and commitment, members of the NETCARE Scientific Steering Committee, and other NETCARE collaborators.

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



| Date | Location(s) | Platform |
|------|-------------|----------|
| 2014 (July – August) | Canadian Arctic Archipelago | Ship – CCGS *Amundsen* |
| 2014 (July) | Resolute Bay, Nunavut | Airborne – Polar6 |
| 2014 (March – July) | Alert, Nunavut | Ground – Dr. Neil Trivett Global Atmosphere Watch Observatory |
| 2015 (April) | Longyearbyen, Alert, Eureka, Inuvik | Airborne – Polar6 |
| 2016 (July – August) | Canadian Arctic Archipelago | Ship – CCGS *Amundsen* |
| 2016 (March, June – July) | Alert, Nunavut | Ground – Dr. Neil Trivett Global Atmosphere Watch Observatory |

**Table 1. NETCARE Arctic Field Campaigns**





**Figure 1. Map of the Arctic indicating NETCARE field work locations, including the ground station (Alert), CCGS *Amundsen* ship tracks in the summers of 2014 and 2016, and Polar 6 aircraft flights in summer 2014 (based out of Resolute Bay) and in spring 2015 (based out of Longyearbyen, Alert, Eureka, and Inuvik).**



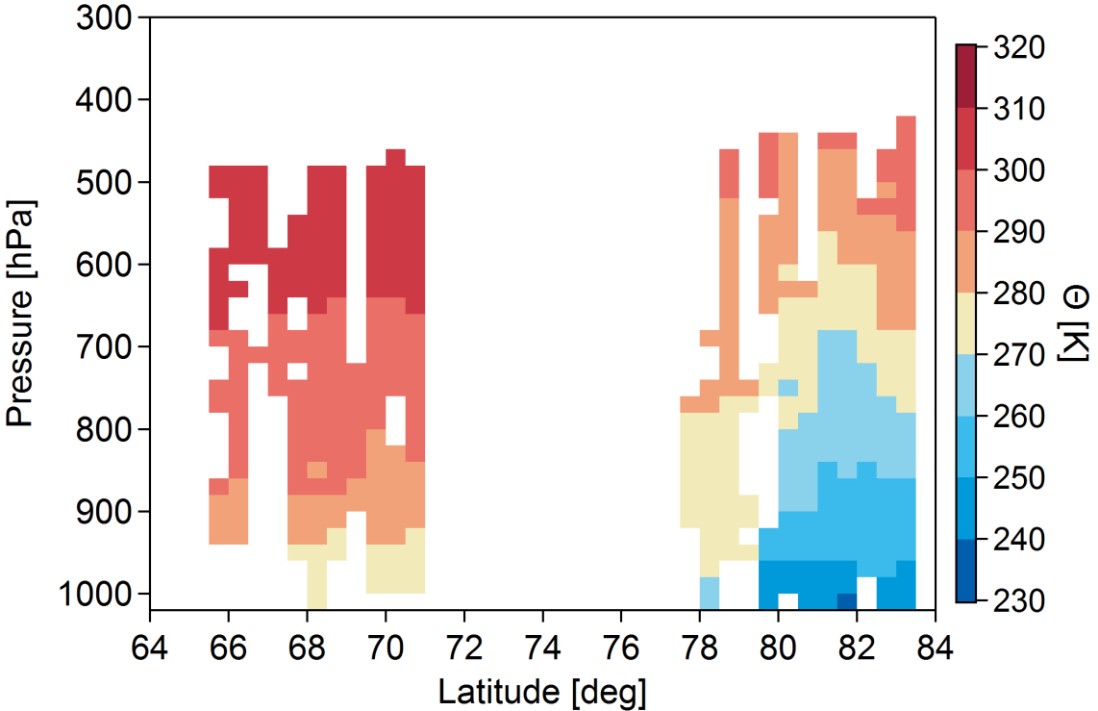

**Figure 2. The potential temperature (Θ) distribution binned in steps of 1° latitude and 20 hPa pressure. Θ was calculated from the temperature and pressure measurements on board the Polar 6 aircraft during the NETCARE 2015 springtime campaign. Minimum potential temperatures of less than 270 K were only observed in the high Arctic lower troposphere, representing very cold air masses that isolate this area from mid-latitudinal influence. The polar dome is formed by the sloping isentropes which can be identified from the NETCARE measurements. Figure from (Bozem et al., 2018).**




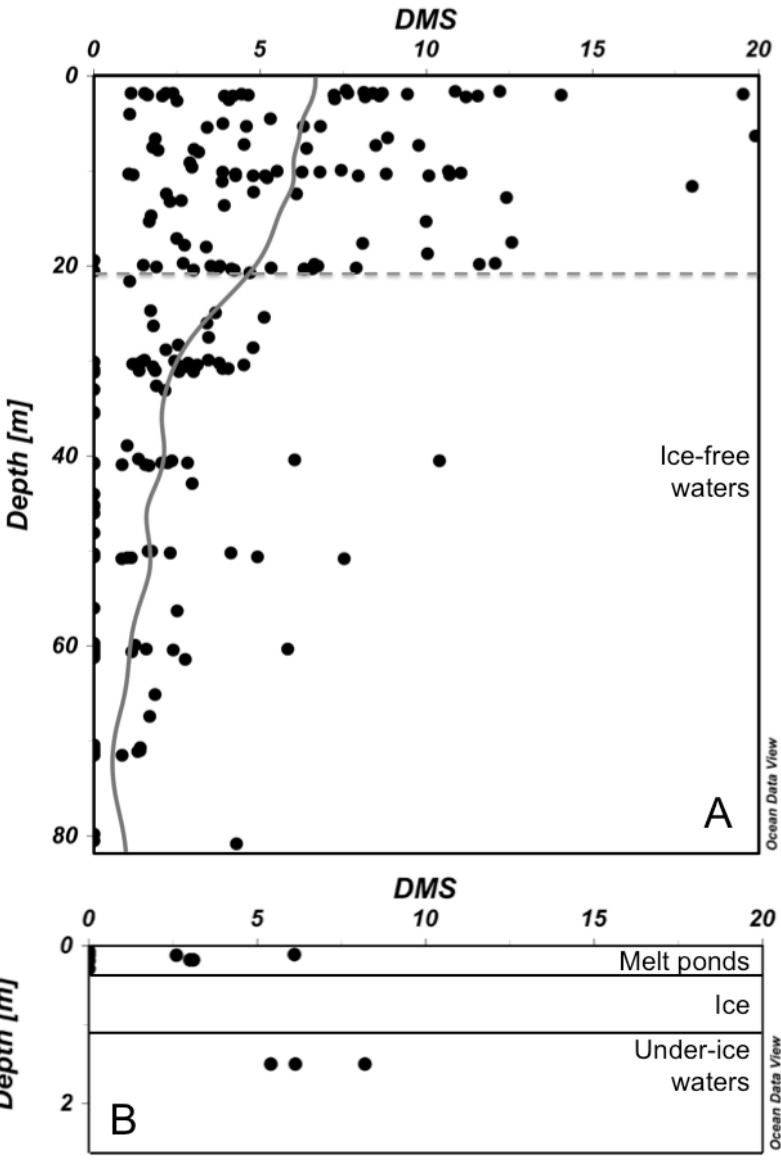

**Figure 3. A) Concentrations of DMS (nmol L⁻¹) in ice-free waters as a function of depth (m) with moving average line (all data, n = 208). The grey dotted line represents the average surface mixed layer depth ($Z_m$ = 21 m) estimated as the depth at which the gradient in density between two successive depths was > 0.03 kg m³. B) Concentrations of DMS (nmol L⁻¹) in melt ponds (n = 9) atop first-year sea ice (Gourdal et al., 2018) and in under-ice waters (n = 3). All data from the 2014 NETCARE cruise onboard CCGS Amundsen.**





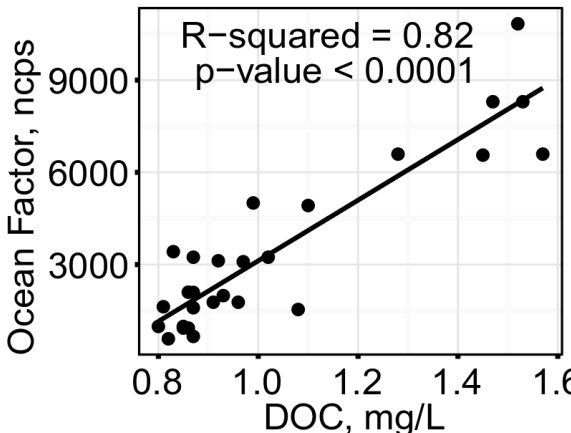
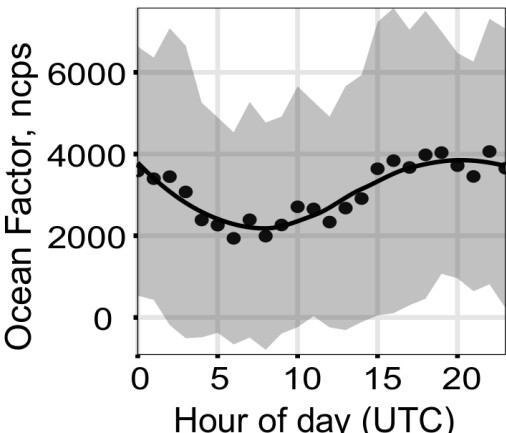

**Figure 4. A large suite of oxygenated VOCs (OVOCs) were measured on the CCGS *Amundsen* during the 2014 cruise in the high Canadian Arctic. A factor analysis of the full time-dependent data set yielded an "Ocean Factor" of small organic acids whose intensity correlated with the DOC levels in the ocean (see left frame) and with time of day and hence downwelling radiation (right frame). See text for additional discussion. Figures from (Mungall et al., 2017).**





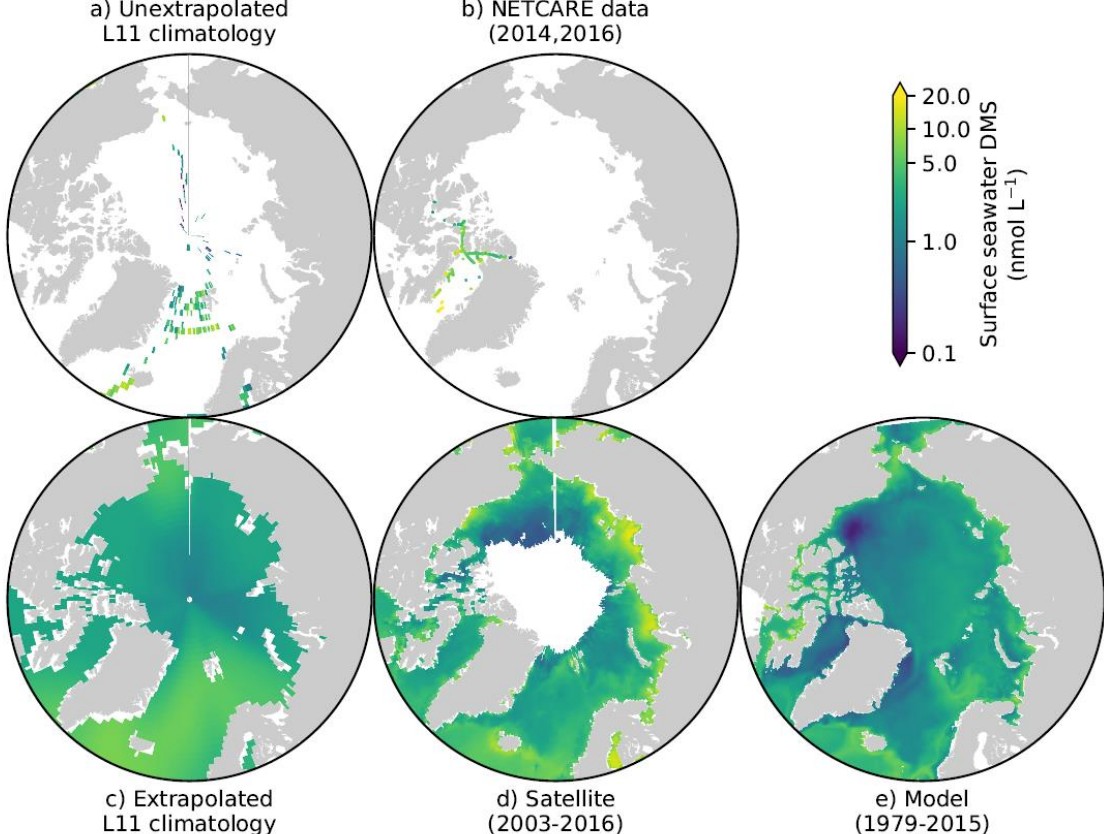

**Figure 5. Pan-Arctic distribution of July-August concentrations of surface ocean DMS. Upper panel shows the comparison between a) the discrete (Lana et al., 2011) climatology and b) the data collected during the two NETCARE field campaigns. Lower panel compares c) the standard (Lana et al., 2011) climatology with d) the satellite-derived (Galí et al., 2018b; Hayashida et al., 2018b) and e) model-based (Hayashida et al., 2018b) climatologies developed within NETCARE.**





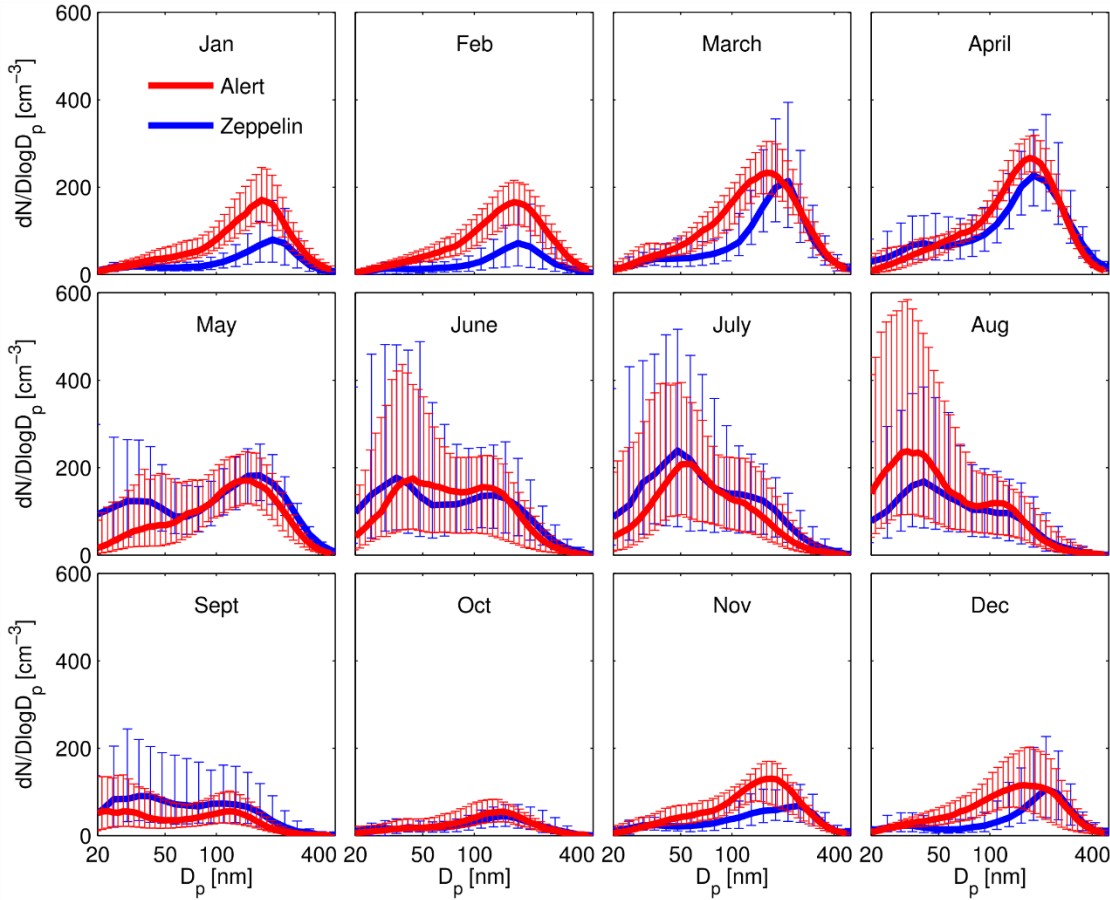

**Figure 6.** Aerosol size distributions from Alert and Zeppelin Arctic field stations. The pronounced accumulation mode in the winter and spring is characteristic of Arctic haze. The mode of Aitken particles is a common feature of the Arctic summertime atmosphere. Figure from (Croft et al., 2016b).





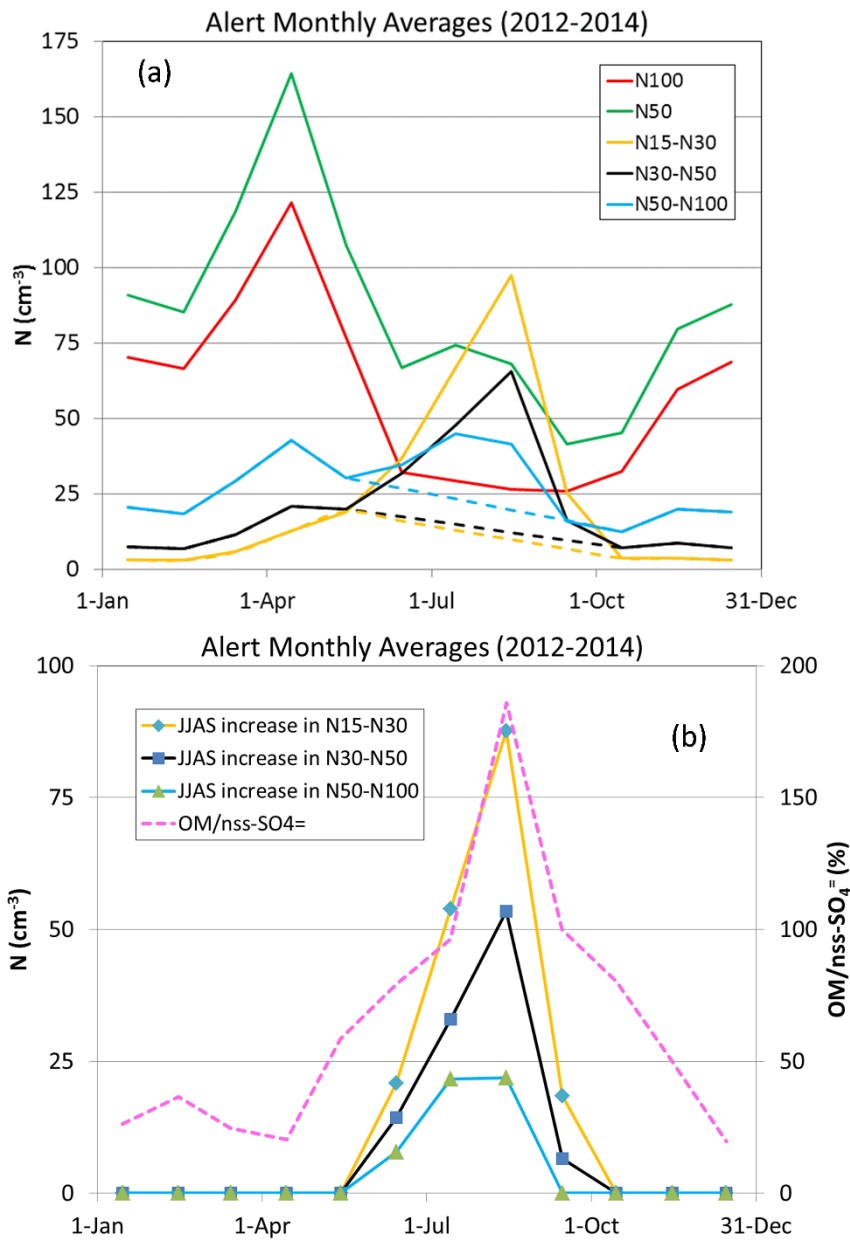

**Figure 7. The changing composition and size distributions of aerosol in the high Arctic. See Supplementary Information for details. a) Monthly average number concentrations for the indicated size ranges for measurements at Alert. b) Estimated increases in particles in the indicated size intervals for June–September, inclusive, and monthly average values of OM/nss-SO$_4^{2-}$ based on weekly filter samples. The data presented here are from April 2012 to October 2014, inclusive.**



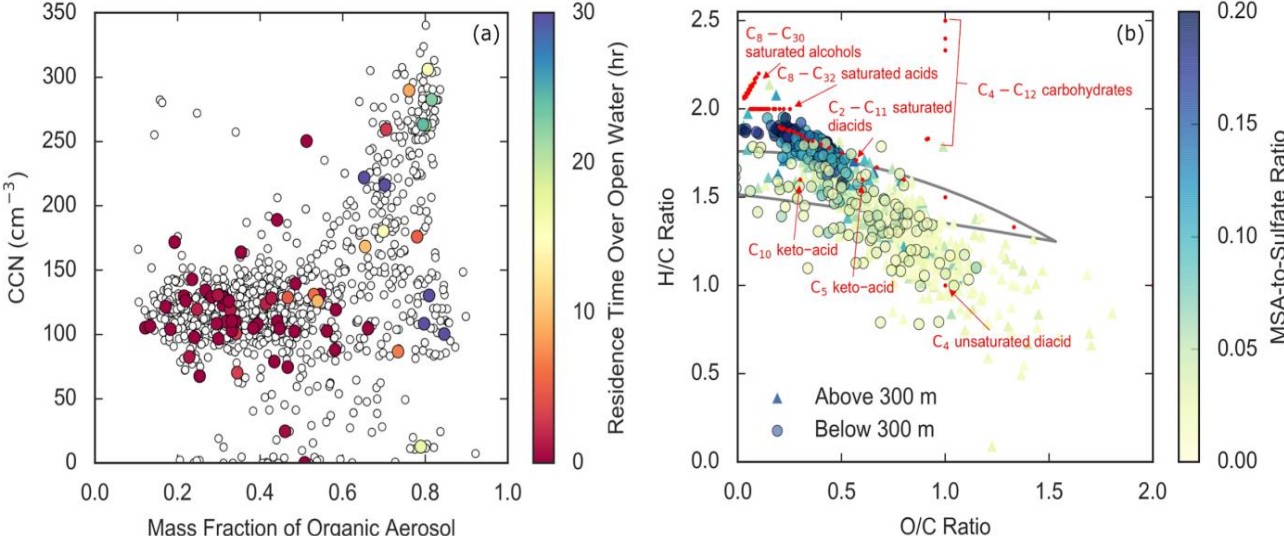

**Figure 8. Panel (a) illustrates that the number of CCN (at 0.6 % supersaturation) measured by NETCARE in the summertime Arctic in 2014 is related to the organic mass fraction of the particles measured by an aerosol mass spectrometer. Open circles are all the data points. The closed, coloured circles represent the FLEXPART-WRF predicted air mass residence time over open water in the boundary layer prior to the measurement (see (Willis et al., 2017) for details). Panel (b) plots the H/C vs. the O/C ratios of submicron aerosol measured during the same summertime 2014 campaign. The circles and triangles are low (<300 m) and high (>300 m) altitude points, and the colour is the MSA-to-sulfate ratio of the aerosol. High ratios indicate large biogenic secondary impact. The convergence of points with high ratios to an H/C ratio close to 2 indicates a composition with substantial hydrocarbon-like character, as indicated in red by the placements for common molecules. Figures from (Willis et al., 2017).**





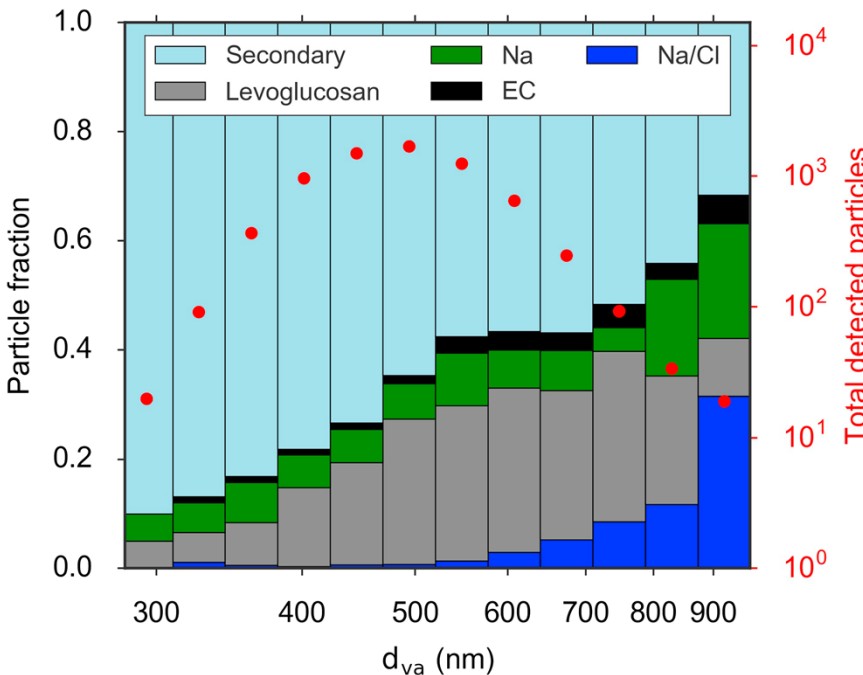

**Figure 9.** **Single particle mass spectrometry results from the NETCARE 2014 summer campaign, where the detected particle fraction is plotted against the aerodynamic diameter of the particle. The total number of particles detected in a specific size bin is plotted in red. The classifications of particle types containing different species are: Na/Cl (dark blue), levoglucosan (grey), Na (green), elemental carbon (EC) (black) and a category of particles called Secondary that include organics, potassium, sulfate, trimethylamine and MSA (light blue). Figure from (Willis et al., 2017).**



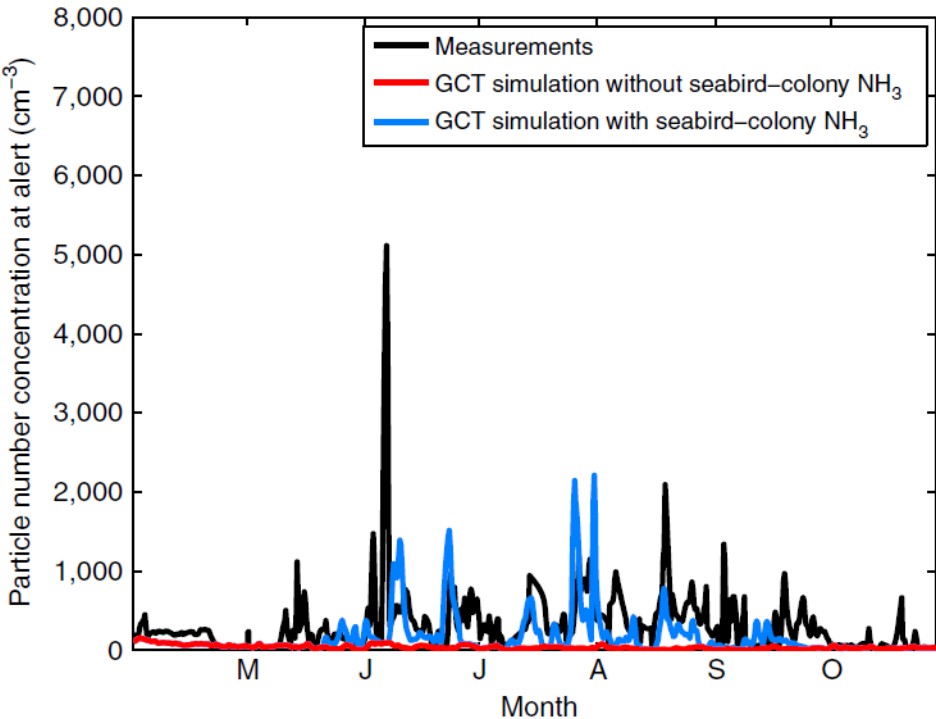

**Figure 10. Time series of measured and modelled numbers of particles 10 nm and larger at Alert during 2011. Sea-bird ammonia is included in the blue curve simulation but not in the red curve simulation. Measurements are in black. Figure from (Croft et al., 2016a).**





**Figure 11. GEOS-Chem adjoint modelling results for BC sources to the Arctic. Panels (a) and (b) show in colour the contributions of BC from different anthropogenic emission and biomass burning regions to the vertical profiles in the atmosphere, where the measurements are in black. Modelled results are for the entire Arctic for the annual average. The data are binned in pressure ranges in panels (c) and (d). Numbers of measurements are along the y axis. Figure from (Xu et al., 2017).**



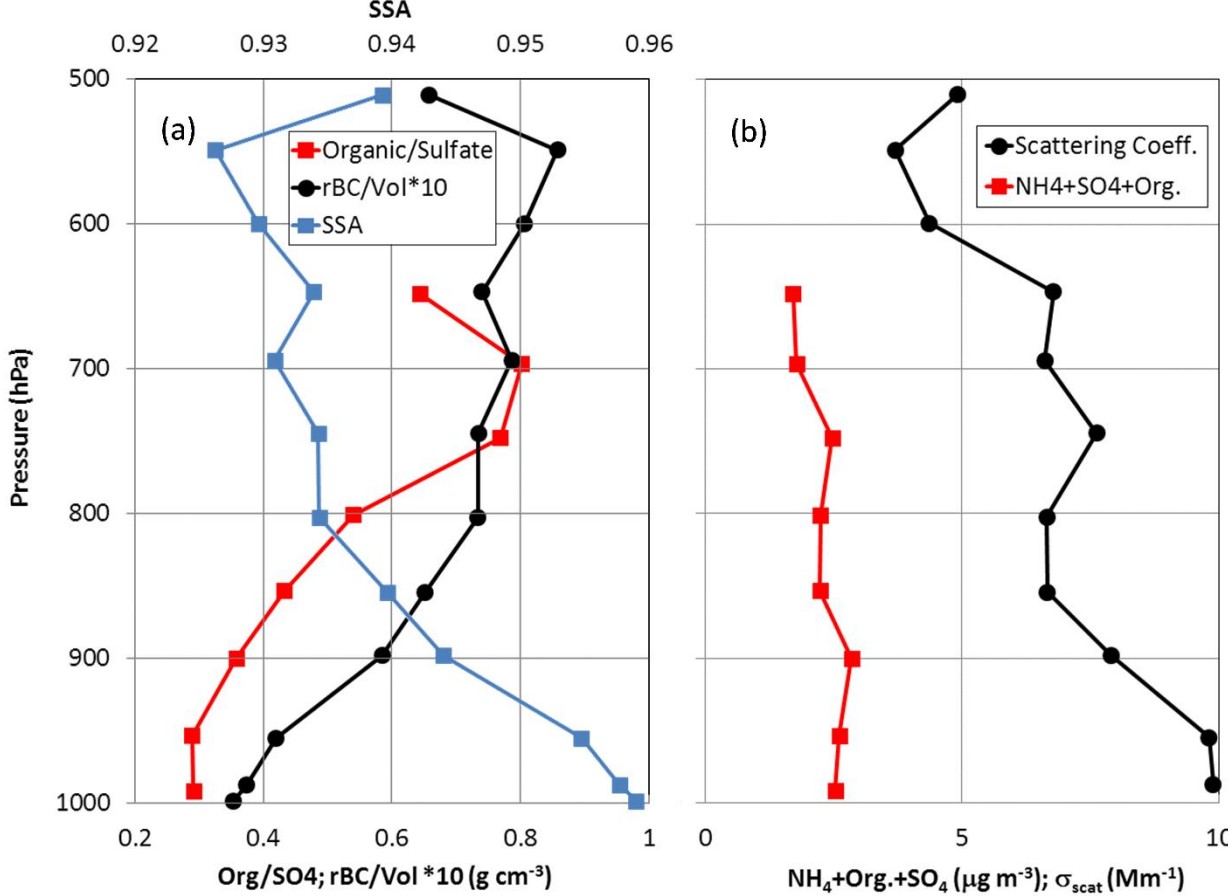

**Figure 12. a) Vertical profiles of the ratio of organic material to sulfate (Organic/Sulfate) from Willis et al. (2018), the ratio of refractory black carbon (rBC) to the volume concentration of the submicron aerosol estimated from the measured size distribution (Schulz et al., 2018) and the aerosol single scatter albedo (Leaitch et al., 2018a). b) Profiles of the sum of the mass concentrations of ammonium (NH4), organic material (Org.) and sulfate (SO4) with the light scattering coefficient ($\sigma_{scat}$). All values are medians over approximately 50 hPa pressure intervals. Results are for flights conducted out of Alert and Eureka, and constrained to $\sigma_{scat} <15$ Mm[-1], which represents 98 % of the observed $\sigma_{scat}$.**



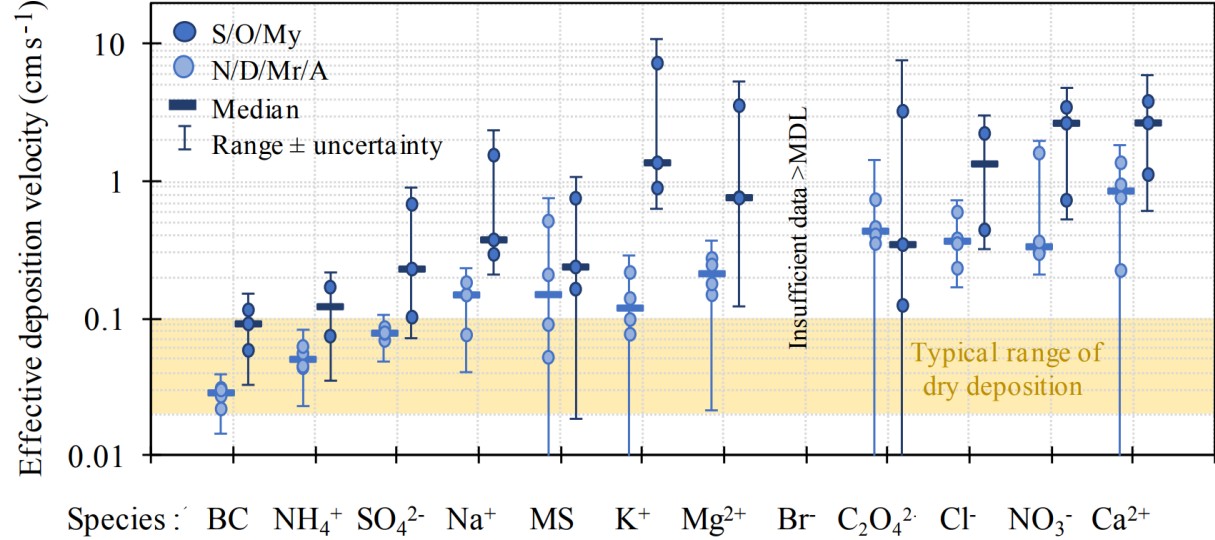

**Figure 13.** The monthly average (circles) effective deposition velocity of different chemical species to snow at Alert during 2014-2015. Median values (bars) are also shown. The effective deposition velocity encompasses both wet and dry deposition processes. In general, the warmer months have higher deposition velocities than the colder months, likely due to enhanced wet deposition in the former. Figure from (Macdonald et al., 2017).



**Figure 14. Results from measurements in the Arctic marine boundary layer during summer 2014 (Irish et al., 2018a). A) Ratios of the surface area of mineral dust particles to the surface area of sea salt particles measured by computer controlled scanning electron microscopy with energy dispersion X-ray spectroscopy (CCSEM-EDX). Ratios of predicted INP concentrations from mineral dust, [INP(T)]$_{MD}$, to the predicted INP concentrations from sea spray aerosol, [INP(T)]$_{SS}$, calculated using CCSEM-EDX measurements at temperatures of B) −25°C, C) −20°C, and D) −15°C. Results show that mineral dust is a more important contributor to the INP population in the Arctic than sea spray aerosol at these times and locations.**