# Peer review of "New insights into aerosol and climate in the Arctic"

_Atmospheric Chemistry and Physics, 2018_

## Referee Comment (RC1) · Anonymous Referee #1 · 30 Oct 2018

This is an overview paper that summarizes key findings from the large NETCARE research campaign focused on aerosols and clouds in the Arctic. The paper is slightly unusual in that it only lightly touches on a wide variety of research findings that are described in other NETCARE papers, but I find synthesis papers of this nature to be useful because they serve as a starting point for informing on the campaign and as a gateway for readers to locate more detailed studies and place them in the broader context of the campaign and outstanding research questions. As a standalone, this paper is also useful in highlighting recent advances in Arctic aerosol research and key outstanding questions that persist today. The paper is clearly written and organized. I recommend publication after minor issues are addressed.

One statement that requires investigation, however, is on p.17, line 1: "GEOS-Chem-

[Figure]

Tomas yields a pan-Arctic average springtime DRE ranging from -1.65 W/m2 for entirely externally mixed BC to -1.34 W/m2 for entirely internally mixed BC." - The top-of-atmosphere DRE of Arctic BC is most certainly positive (See, e.g., Table 1 of Samset et al 2013, doi:10.5194/acp-13-2423-2013, showing positive global and Arctic BC DRE from all models). Do the cited estimates perhaps refer to DRE by all aerosols? Or do the numbers perhaps include indirect BC effects that are negative? Please clarify. If the DRE estimate is for all aerosols, please communicate which anthropogenic and natural aerosol groups are included in the estimate. It would also be helpful to include the isolated DRE of BC if possible, since BC is the focus of this paragraph.

Minor issues:

Abstract: It would be helpful to see more concrete or quantitative findings presented in the abstract, where possible. In particular: (1) line 24: "a significant fraction of the new particles grow..." - What was the actual (range of) fraction that was found? Or... how 'significant' is this fraction? (2) line 30-31: "... measurements were used to better establish the BC source regions that supply the Arctic..." - And which source regions were found to be important? Was there a change in our general understanding of the important source regions?

p.4, line 3: References cited in this manner should, I believe, be: "Quinn et al (2006)" instead of "(Quinn et al, 2006)".

p.8, line 13: Please provide a reference or link for "PMEL database".

p.8, line 25: Why are the DMS dynamics so different between multi-year and first-year ice? Even speculation on this would be useful.

p.9, line 14: "Biogenic DMS oxidation products were also prevalent in the marine boundary layer" - The relevance of this statement is not immediately clear. It would be helpful to connect it better with the rest of the paragraph.

p.10, line 14: "... tundra could act as a source of ammonia..." - The abstract and conclusions highlight sea birds as an important and underappreciated source of ammonia. It would be helpful if you can link the different NETCARE studies together to draw conclusions on the relative importance of the tundra soil and sea birds as sources of ammonia in different environments and/or seasons.

p.11, line 11: "Furthermore, the simulated response of the mean cloud radiative forcing in the Arctic is proportional to the mean surface seawater DMS concentration in the Arctic" - Do you have any explanation for why? It is not immediately clear to me why this should be the case, but perhaps it is intuitive and I am not reading it correctly.

p.12, line 19: Regarding the relative importance of European and Asian sources to Arctic BC at different altitudes: One study that has explored this via modeling is Jiao et al (2016, doi:10.1002/2015JD023964).

p.15, line 16: Why is the dome boundary ("north of 66-68.5") expressed as a range? Does the boundary vary with longitude in your analysis? With time?

p.15, line 29: What is the meaning of "sensitivity to the surface"?

p.16, line 9-10: This sentence is a bit unclear. By "changes", do you mean seasonal changes? And by "time period" do you you mean winter to spring? Please clarify.

p.19, line 8: "... still debated in the literature" - Please provide references that communicate this debate.

Figure 7: Please explain this figure more thoroughly, including the legend description and meaning of dashed lines.

---

## Referee Comment (RC2) · Anonymous Referee #2 · 29 Nov 2018

This manuscript provides a detailed overview of the science highlights that have arisen from the Canadian NETCARE programme. It serves as an access portal to the extensive collection of literature now available from analysis of NETCARE data and modelling activities, including several special issue journal collections. The paper aims to synthesise the NETCARE results, bringing them together into three broad themes, and does a nice job in illustrating where NETCARE has brought about new understanding in Arctic aerosol research, as well as identifying remaining open questions. The paper is very well written and presented, is very accessible to the general atmospheric and climate science communities, and its content is well within the remit of ACP. I recommend that the paper be published, subject to addressing the following minor comments.

There is a lot of dense information presented. It may be a challenge for the reader to

efficiently find a specific aspect that may be of particular interest. One idea to help this would be something like a contents table up front, perhaps at the end of the Introduction section, listing the topics presented in each sub-section of Sections 3-5. However, I am not sure how this would sit with ACP editorial policy.

Line 5: The link between long-range transport and Arctic environment / climate is perhaps not so obvious without explicit mention of the role of long-range transport in controlling aerosol and trace gas sources.

Line 7: "Arctic amplification of radiative forcing" Do Arctic feedbacks strictly amplify a forcing, or the temperature / climate response to a forcing?

Page 6, Line 3/4: Do changes in transport patterns also affect the winter / summer differences in aerosol loading?

Page 6, line 6: "dome keeps the summertime Arctic nearly free of anthropogenic aerosol" Is this the case generally, or just at the surface? Data from the POLARCAT studies in 2008 demonstrates some influence from remote sources aloft in summer (e.g. Schmale et al., 2011).

Page 7, line 28: A reference to Table 1 / Figure 1 would be useful when referring to the NETCARE summer campaigns.

Page 9, line 22-25: Potential source of highly water soluble oVOCs. Is there any information that can help elucidate the more likely mechanism? What specifically are the compounds that could lead to SOA? Is there evidence for these having a common source with those measured in the Mungall study?

Page 10, Line 19 - Mention of Lana climatology. Perhaps some context needed here? I felt that this statement came out of the blue. Perhaps a reiteration / statement that this is standard dataset used in models, with appropriate references is needed. Or perhaps this statement could be made more explicitly earlier in Section 3.

Page 14, line 1: "Natural emissions of ammonia are also important in this context." As

this is a new paragraph, "this context" is unclear. Please clarify.

Section 4.2 - discussion of ammonia sources. Fires, tundra and seabirds are all mentioned. It would be useful for the reader to know the relative magnitudes of these sources in terms of their Arctic influences. Is information available e.g. from the GEOS-CHEM modelling study?

Page 16, line 24 - In what sense is the GEOS-CHEM adjoint source contribution study different from previous studies? Simply in terms of improved spatial information on sources? Different conclusions on importance of source regions? Or something else?

Page 18, line 25: Please provide a reference for the assertion regarding expected future increases in Arctic shipping.

Page 25, line 6: ".. transport of long-range pollutants.." Not sure what would be defined as a "long-range pollutant". Should this be "long-range transport of pollutants.."?

Page 25, line 24: Since the IMPAACT project is not yet funded, it might be helpful to explicitly mention that IMPAACT is part of a broader activity aimed at reducing uncertainties in pollution processing during LRT to the Arctic under the PACES umbrella (https://pacesproject.org/about).

References Schmale, J., J. et al., (2011), Source Identification and Airborne Chemical Characterisation of Aerosol Pollution from Long-range Transport over Greenland during POLARCAT Summer campaign 2008, Atmos. Chem. Phys., 11, 10097-10123.

---

## Author Comment (AC1) · 1 Feb 2019

This is an overview paper that summarizes key findings from the large NETCARE research campaign focused on aerosols and clouds in the Arctic. The paper is slightly unusual in that it only lightly touches on a wide variety of research findings that are described in other NETCARE papers, but I find synthesis papers of this nature to be useful because they serve as a starting point for informing on the campaign and as a gateway for readers to locate more detailed studies and place them in the broader context of the campaign and outstanding research questions. As a standalone, this paper is also useful in highlighting recent advances in Arctic aerosol research and key outstanding questions that persist today. The paper is clearly written and organized. I recommend publication after minor issues are addressed.

*Response: Thank you for the useful review.*

One statement that requires investigation, however, is on p.17, line 1: "GEOS-Chem-Tomas yields a pan-Arctic average springtime DRE ranging from -1.65 W/m2 for entirely externally mixed BC to -1.34 W/m2 for entirely internally mixed BC." - The top-of-atmosphere DRE of Arctic BC is most certainly positive (See, e.g., Table 1 of Samset et al 2013, doi:10.5194/acp-13-2423-2013, showing positive global and Arctic BC DRE from all models). Do the cited estimates perhaps refer to DRE by all aerosols? Or do the numbers perhaps include indirect BC effects that are negative? Please clarify. If the DRE estimate is for all aerosols, please communicate which anthropogenic and natural aerosol groups are included in the estimate. It would also be helpful to include the isolated DRE of BC if possible, since BC is the focus of this paragraph.

*Response: The line is indeed referring to the DRE from all aerosols and how the mixing state of BC affects this estimate. To make this more clear, we have changed the sentence to: "GEOS-Chem-TOMAS yields a pan-Arctic average springtime DRE for all aerosols ranging from -1.65 W/m2 when assuming entirely externally mixed BC to -1.34 W/m2 when assuming entirely internally mixed BC." (page 18, line 27), i.e. this is for all aerosols (natural and anthropogenic) versus no aerosol.*

*Regarding the comment "It would also be helpful to include the isolated DRE of BC if possible, since BC is the focus of this paragraph.", species-isolated direct effects are not well defined for the core-shell internal mixtures because, for example, the amount of coating impacts the BC absorption. This is why we didn't include these values in the paper.*

Minor issues:
Abstract: It would be helpful to see more concrete or quantitative findings presented in the abstract, where possible. In particular: (1) line 24: "a significant fraction of the new particles grow..." - What was the actual (range of) fraction that was found? Or... how 'significant' is this fraction? (2) line 30-31: "... measurements were used to better establish the BC source regions that supply the Arctic..." - And which source regions were found to be important? Was there a change in our general understanding of the important source regions?

*Response:*

*Thanks for this comment. We have tried to make the Abstract more specific by adding the following text:*

*With respect to DMS: "Unexpectedly high summertime dimethyl sulfide (DMS) levels were identified in ocean water (up to 75 nM) and the overlying atmosphere (up to 1 ppbv) in the Canadian Arctic Archipelago (CAA). Furthermore, melt ponds, which are widely prevalent, were identified as an important DMS source (with DMS concentrations of up to 6 nM and a potential contribution to atmospheric DMS of 20% in the study area)." (page 2, line 18)*

*With respect to nucleation and growth: "Evidence was found of widespread particle nucleation and growth in the marine boundary layer in the CAA in the summertime, with these events observed on 41% of days in a 2016 cruise. As well, at Alert, Nunavut particles that are newly formed and grown under conditions of minimal anthropogenic influence during the months of July and August are estimated to contribute 20 to 80% of the 30-50 nm particle number density." (page 2, line 22; also, new text was added at page 14, line 28 and in the last paragraph of the Supplementary Information supporting this statement)*

*With respect to black carbon sources: "… with evidence for a dominant springtime contribution from eastern and southern Asia to the middle troposphere, and a major contribution from northern Asia to the surface." (page 2, line 33)*

*With respect to black carbon sources: "Amongst multiple aerosol components, BC was observed to have the smallest effective deposition velocities to high Arctic snow (0.03 cm/s)." (page 3, line 5)*

p.4, line 3: References cited in this manner should, I believe, be: "Quinn et al (2006)" instead of "(Quinn et al, 2006)".

*Response: This change has been made.*

p.8, line 13: Please provide a reference or link for "PMEL database".

*Response:  The text has been changed to remove the PMEL database reference:  "These concentrations are higher than the area-weighted mean of ca. 2.4 nmol L$^{-1}$ derived from the global climatology of Lana et al. (2011), bringing support to the suggestion that melt ponds may represent a significant source of DMS in the Arctic." (page 9, line 21)*

p.8, line 25: Why are the DMS dynamics so different between multi-year and first-year ice? Even speculation on this would be useful.

*Response: The potential reasons why DMS dynamics are different between the multi-year and first-year ice system are explored in the phrases following line 6 of page 10, to the end of the section.*

*However, to enhance clarity, a phrase invoking the hypothesis behind the contrasting dynamics was added: "Contrasting DMS dynamics between FYI and MYI systems were likely linked to differences in light penetration through the ice pack and its availability to primary producers in the waters just below the ice." (page 10, line 3)*

p.9, line 14: "Biogenic DMS oxidation products were also prevalent in the marine boundary layer" - The relevance of this statement is not immediately clear. It would be helpful to connect it better with the rest of the paragraph.

*Response: This sentence has been changed to: "Evidence for atmospheric DMS was the widespread prevalence of biogenic DMS oxidation products in the marine boundary layer (Ghahremaninezhad et al., 2016)." (page 10, line 26)*

p.10, line 14: "... tundra could act as a source of ammonia..." - The abstract and conclusions highlight sea birds as an important and underappreciated source of ammonia. It would be helpful if you can link the different NETCARE studies together to draw conclusions on the relative importance of the tundra soil and sea birds as sources of ammonia in different environments and/or seasons.

*Response: Measurements at the Alert site in 2016 indicated that the local tundra could be a net source of ammonia to the Arctic atmosphere, but the measurements were not extensive enough to allow for a more general assessment. In Croft et al., ACPD 2018, the potential importance of tundra emissions is evaluated through a modelling scenario in which the upper end of inferred tundra NH3 emissions (2.2 $ng$ $m^{-2}$ $s^{-1}$) is included in the model, and the impact on atmospheric NHx is comparable to that of the source from seabirds. Work to evaluate whether emissions from the tundra across the Arctic can be this important is ongoing.*

p.11, line 11: "Furthermore, the simulated response of the mean cloud radiative forcing in the Arctic is proportional to the mean surface seawater DMS concentration in the Arctic" - Do you have any explanation for why? It is not immediately clear to me why this should be the case, but perhaps it is intuitive and I am not reading it correctly.

*Response:  In the revised version of the manuscript, this sentence now reads: "Furthermore, the simulated response of the mean cloud radiative forcing in the Arctic is approximately proportional to the mean surface seawater DMS concentration in the Arctic". (page 13, line 9)*

*This conclusion is based on model results summarized in Fig. 8 in Mahmood et al. (2018). Climate model simulations were conducted using specified SSTs and sea ice. The simulations produce only minor changes in simulated near-surface winds in response to different surface seawater DMS concentration data sets. Furthermore, the model produces a linear response in mean DMS emissions and atmospheric DMS concentrations to changes in mean surface seawater DMS concentrations in the Arctic (i.e. at latitudes > 60N), to a very good approximation. In addition, simulated sulfate aerosol deposition rates and cloud radiative forcings are linearly related to sulfate concentrations, to good approximation. However, the natural variability in cloud properties is considerable and the statistical significance of the results is low given the relatively small ensemble of simulations in the study. Tesdal et al. (www.atmos-chem-phys.net/16/10847/2016/) also find a nearly linear relationship between global mean cloud radiative forcing and DMS emissions in simulations with a bulk aerosol model, which agrees with our conclusions for the Arctic. Details are provided in Mahmood et al. (2018).*

p.12, line 19: Regarding the relative importance of European and Asian sources to Arctic BC at different altitudes: One study that has explored this via modeling is Jiao et al (2016, doi:10.1002/2015JD023964).

*Response: This reference has been added.*

p.15, line 16: Why is the dome boundary ("north of 66-68.5") expressed as a range? Does the boundary vary with longitude in your analysis? With time?

*Response: Bozem et al. (2019) calculate isentropic trace gas gradients in layers of 2 K for the potential temperature as the vertical coordinate to derive the latitude of the polar dome boundary over all longitudes that were covered by the measurements. Hence, for every 2 K altitude interval they determined the latitude of the strongest trace gas gradient. They finally used the median of these maximum gradient latitudes to define a dome boundary. If a different median value for the maximum gradients for the two species CO and CO2 was derived, they consider this difference as the range of the polar dome boundary which can in turn be interpreted as a transition zone rather than a sharp boundary between inside and outside of the polar dome. In this analysis the polar dome boundary was calculated as an average over the airborne campaign period (5.4. – 21.4.2015). We emphasize that the polar dome is largely a conceptual idea.*

p.15, line 29: What is the meaning of "sensitivity to the surface"?

*Response: Our intended meaning is that air masses present at higher altitudes and higher potential temperatures were more recently influenced by lower latitude sources at the surface. This sentence has been revised as follows: "Air masses at lower potential temperature (lower altitude) spent long times (> 10 days) in the polar dome, while air masses at higher potential temperature (higher altitude) had entered the Arctic more recently and were more recently influenced by lower latitude sources at the surface (Willis et al., 2019)." (page 17, line 19)*

p.16, line 9-10: This sentence is a bit unclear. By "changes", do you mean seasonal changes? And by "time period" do you you mean winter to spring? Please clarify.

*Response: The revised version of the manuscript now includes the following sentence: "Overall, sources and sinks of BC in the Arctic are well balanced, leading to nearly steady Arctic burdens during the time period from December to May." (page 17, line 33)*

p.19, line 8: "... still debated in the literature" - Please provide references that communicate this debate.

*Response: We have added references that communicate this debate. The manuscript has been edited as follows: "the effect of engine load on BC emission factors is still debated in the literature. While various authors report increasing emission factors by decreasing engine loading (Agrawal et al., 2008; Petzold et al. 2010, 2011; Khan et al., 2012), other authors report decreasing emission factors by decreasing engine loading (Cappa et al., 2014)." (page 21, line 2)*

*Agrawal, H., Welch, W. A., Miller, J. W., and Cocker, D. R.: Emission measurements from a crude oil tanker at sea, Environ. Sci. Technol., 42, 7098–7103, doi:10.1021/es703102y, 2008.*

*Petzold, A., Weingartner, E., Hasselbach, J., Lauer, P., Kurok, C., and Fleischer, F.: Physical properties, chemical composition, and cloud forming potential for particulate emissions from a marine diesel engine at various load conditions, Environ. Sci. Technol., 44, 3800–3805, doi:10.1021/es903681z, 2010.*

*Petzold, A., Lauer, P., Fritsche, U., Hasselbach, J., Lichtenstern, M., Schlager, H., and Fleischer, F.: Operation of marine diesel engines on biogenic fuels: modification of emissions and resulting climate effects, Environ. Sci. Technol., 45, 10394–10400, doi:10.1021/es2021439, 2011.*

*Khan, M. Y., Giordano, M., Gutierrez, J., Welch, W. A., Asa-Awuku, A., Miller, J. W., and Cocker, D. R.: Benefits of two mitigation strategies for container vessels: cleaner engines and cleaner fuels, Environ. Sci. Technol., 46, 5049–5056, doi:10.1021/es2043646, 2012.*

*Cappa, C. D., Williams, E. J., Lack, D. A., Buffaloe, G. M., Coffman, D., Hayden, K. L., Herndon, S. C., Lerner, B. M., Li, S.- M., Massoli, P., McLaren, R., Nuaaman, I., Onasch, T. B., and Quinn, P. K.: A case study into the measurement of ship emissions from plume intercepts of the NOAA ship Miller Freeman, Atmos. Chem. Phys., 14, 1337–1352, doi:10.5194/acp-14-1337-2014, 2014.*

Figure 7: Please explain this figure more thoroughly, including the legend description and meaning of dashed lines.

*Response: The revised caption is:*

*"Figure 7. The changing composition and size distributions of aerosol in the high Arctic. See Supplementary Information for details.  a) Monthly average number concentrations for the indicated size ranges for measurements at Alert. b) Estimated increases in particles in the indicated size intervals for June–September, inclusive, and monthly average values of OM/nss-SO4$^{2-}$ based on weekly filter samples. The data presented here are from April 2012 to October 5 2014, inclusive.  The dashed lines in a) represent an estimate of number concentrations assuming no new particle formation.  The number concentration curves in b) are the difference between the solid and dashed curves in a)."*

**Anonymous Referee 2**

This manuscript provides a detailed overview of the science highlights that have arisen from the Canadian NETCARE programme. It serves as an access portal to the extensive collection of literature now available from analysis of NETCARE data and modelling activities, including several special issue journal collections. The paper aims to synthesise the NETCARE results, bringing them together into three broad themes, and does a nice job in illustrating where NETCARE has brought about new understanding in Arctic aerosol research, as well as identifying remaining open questions. The paper is very well written and presented, is very accessible to the general atmospheric and climate science communities, and its content is well within the remit of ACP. I recommend that the paper be published, subject to addressing the following minor comments. There is a lot of dense information presented. It may be a challenge for the reader to efficiently find a specific aspect that may be of particular interest. One idea to help this would be something like a contents table up front, perhaps at the end of the Introduction section, listing the topics presented in each sub-section of Sections 3-5. However, I am not sure how this would sit with ACP editorial policy.

*Response:  Thank you for the useful review.  The structure of the paper is now listed in full at the end of the Introduction.*

Page 3, Line 5: The link between long-range transport and Arctic environment / climate is perhaps not so obvious without explicit mention of the role of long-range transport in controlling aerosol and trace gas sources.

*Response: The text has been changed to: "Rapid changes in the Arctic environment including rising temperatures, melting sea ice, elongated warm seasons and changing aerosol and trace gas long-range transport patterns …" (page 3, line 8)*

Page 3, Line 7: "Arctic amplification of radiative forcing" Do Arctic feedbacks strictly amplify a forcing, or the temperature / climate response to a forcing?

*Response: Thanks.  The wording has been changed to: " … it is particularly important to understand the feedback processes that lead to amplification of Arctic warming." (page 3, line 11)*

Page 6, Line 3/4: Do changes in transport patterns also affect the winter / summer differences in aerosol loading?

Page 6, line 6: "dome keeps the summertime Arctic nearly free of anthropogenic aerosol" Is this the case generally, or just at the surface? Data from the POLARCAT studies in 2008 demonstrates some influence from remote sources aloft in summer (e.g. Schmale et al., 2011).

*Response: Thanks for these two comments.  To address them, the relevant paragraph has been changed to:*

*"The decline in Arctic haze after its peak in early spring and the approach to the summertime pristine conditions are largely related to changes in transport as the polar front moves northward and aerosol scavenging rather than a reduction in aerosol production. Wet deposition associated with transport across the retracted polar front, frequent low-intensity precipitation and longer residence times within the polar dome keeps the summertime near-surface Arctic nearly free of anthropogenic aerosol (Barrie,*

*1986; Stohl, 2006; Garrett et al., Browse et al., 2012). However, at higher altitudes up to 8 km, long range-transport from mid-latitude pollution into the Arctic was observed also in summer (Schmale et al., 2011). Marine sources have a strong influence on the Arctic summer aerosol near the surface and possibly aloft (Dall´Osto et al., 2017; Korhonen et al., 2008b; Stohl, 2006).” (page 7, line 9)*

Page 7, line 28: A reference to Table 1 / Figure 1 would be useful when referring to the NETCARE summer campaigns.

*Response: The following text has been added: “see Figure 1 and Table 1” (page 9, line 4)*

Page 9, line 22-25: Potential source of highly water soluble oVOCs. Is there any information that can help elucidate the more likely mechanism? What specifically are the compounds that could lead to SOA? Is there evidence for these having a common source with those measured in the Mungall study?

*Response: These are all good questions. At this point, we don't feel there is more information available to determine the precise mechanism by which the OVOCs arise via the sea surface microlayer. We have added a sentence to the paper that mentions that there was a weak positive correlation between particle volume and the OVOC abundances, indicating a potential link between the two: “We note that there was a weak positive correlation between total aerosol volume and the levels of OVOCs observed, indicating a potential link between the processes forming OVOCs and aerosol growth.” (page 11, line 6)*

Page 10, Line 19 - Mention of Lana climatology. Perhaps some context needed here? I felt that this statement came out of the blue. Perhaps a reiteration / statement that this is standard dataset used in models, with appropriate references is needed. Or perhaps this statement could be made more explicitly earlier in Section 3.

*Response: We agree with the reviewer. Although the Lana et al. 2011 climatology is mentioned on page 8 (line 19), the phrase could have been elaborated upon. We propose the modified version of the text:*

 *“Prior to the NETCARE field campaigns, the existing un-extrapolated DMS climatology, averaged over the most productive time of the year (months of July and August), clearly demonstrated the scarcity of surface ocean DMS measurements in the Arctic (Lana et al. 2011). The updated Lana DMS climatology and its precursor (Kettle et al. 1999) have long represented useful tools for oceanic model validation (eg. Kim et al. 2017; Le Clainche et al. 2010; Tesdal et al. 2016) and lack of data over the Canadian Polar Shelf and the Baffin Bay area challenged the representativeness of the standard (extrapolated) version of this climatology for these specific regions (Fig. 5c).” (page 12, line 2)*

*Kettle, A. J., Andreae, M. O., Amouroux, D., Andreae, T.W., Bates,T. S., Berresheim, H., Bingemer, H., Boniforti, R., Curran, M. A. J., DiTullio, G. R., Helas, G., Jones, G. B., Keller, M. D., Kiene, R. P., Leck, C., Levasseur, M., Malin, G., Maspero, M., Matrai, P., McTaggart, A. R., Mihapoulos, N., Nguyen, B. C., Novo, A., Putaud, J. P., Rapsomanikis, S., Roberts, G., Schebeske, G., Sharma, S., Simó, R., Staubes, R., Turner, S., and Uher, G.: A global database of sea surface dimethylsulfide (DMS) measurements and a procedure to predict sea surface DMS as a function of latitude, longitude, and month, Global Biogeochem. Cy., 13, 399–444, doi:10.1029/1999GB900004, 1999.*

*Le Clainche, Y., Vézina, A., Levasseur, M., Cropp, R. A., Gunson, J. R., Vallina, S. M., Vogt, M., Lancelot, C., Allen, J. I., Archer, S. D., Bopp, L., Deal, C., Elliott, S., Jin, M., Malin, G., Schoemann, V., Simó, R., Six, K. D.,*

*Stefels, J. : A first appraisal of prognostic ocean DMS models and prospects for their use in climate models, Global Biogeochem. Cycles, 24, GB3021, doi:10.1029/2009GB003721, 2010.*

*Tesdal, J.-E., Christian, J. R., Monahan, A. H., and von Salzen, K.: Sensitivity of modelled sulfate aerosol and its radiative effect on climate to ocean DMS concentration and air–sea flux, Atmos. Chem. Phys., 16, 10847-10864, https://doi.org/10.5194/acp-16-10847-2016, 2016.*

*Kim M.J., Novak, G.A., Zoerb, M.C., Yang, M., Blomquist, B.W., Huebert, B.J., Cappa, C.D., Bertram, T.H.: Air-sea exchange of volatile organic compounds and the impact on aerosol size distributions, Geophys. Res. Lett., 44, 3887-3896, doi:10.1002/2017GL072975, 2017.*

Page 14, line 1: "Natural emissions of ammonia are also important in this context." As this is a new paragraph, "this context" is unclear. Please clarify.

*Response: This sentence has been changed to: "Natural emissions of ammonia are also important to new particle formation and growth." (page 15, line 21)*

Section 4.2 - discussion of ammonia sources. Fires, tundra and seabirds are all mentioned. It would be useful for the reader to know the relative magnitudes of these sources in terms of their Arctic influences. Is information available e.g. from the GEOSCHEM modelling study?

*Response: The text has been re-phrased to: "Wentworth et al. (2016) used GEOS-Chem model simulations to interpret NETCARE ammonia measurements (see Section 3.3) and found that migratory seabird colonies (emitting 36 Gg $NH_3$ between May and September) were important sources of ammonia in the summertime Arctic." (page 15, line 23)*

*and*

*"Other natural ammonia sources within the same order of magnitude as the seabird-colony emissions, including but not limited to episodic biomass burning influences (Lutsch et al., 2016; Lutsch et al. (in prep.)) and tundra emissions (Murphy et al., 2018), could also contribute to these effects (Croft et al., 2018)." (page 15, line 32)*

Page 16, line 24 - In what sense is the GEOS-CHEM adjoint source contribution study different from previous studies? Simply in terms of improved spatial information on sources? Different conclusions on importance of source regions? Or something else?

*Response: The analysis of NETCARE measurements contributed to different conclusions on the importance of source regions, while the use of the adjoint tool enabled improved spatial information on sources. We revised the text to the following to indicate these findings:*

*"This dominant role of Asian sources is consistent with some recent studies (e.g. Ikeda et al. 2017; Ma et al. 2013; Wang et al. 2014) but differs from many earlier studies (e.g. Bourgeois and Bey, 2011; Gong et al. 2010; Huang et al. 2010; Shindell et al. 2008; Sharma et al. 2013; Stohl, 2006) due to decreased*

*European emissions and increased Asian emissions. The adjoint simulations enabled identification of pronounced spatial heterogeneity in the contribution of emissions to the Arctic BC column concentrations, with noteworthy contributions from emissions in eastern China (15 %) and western Siberia (6.5 %). The Tarim oilfiled in western China stood out as a specific influential source with an annual contribution of 2.6%. Emissions from as far away as the Indo-Gangetic Plain could have a substantial influence (6.3%) on Arctic BC as well." (page 18, line 14)*

*New references:*

*Gong, S. L., Zhao, T. L., Sharma, S., Toom-Sauntry, D., Lavoué, D., Zhang, X. B., Leaitch, W. R., and Barrie, L. A.: Identification of trends and interannual variability of sulfate and black carbon in the Canadian High Arctic: 1981–2007, J. Geophys. Res., 115, D07305, https://doi.org/10.1029/2009JD012943, 2010.*

*Huang, L., Gong, S. L., Jia, C. Q., and Lavoué, D.: Relative contributions of anthropogenic emissions to black carbon aerosol in the Arctic, J. Geophys. Res., 115, D19208, https://doi.org/10.1029/2009JD013592, 2010.*

*Ikeda, K., Tanimoto, H., Sugita, T., Akiyoshi, H., Kanaya, Y., Zhu, C., and Taketani, F.: Tagged tracer simulations of black carbon in the Arctic: transport, source contributions, and budget, Atmos. Chem. Phys., 17, 10515–10533, https://doi.org/10.5194/acp-17- 10515-2017, 2017.*

*Ma, P.-L., Rasch, P. J., Wang, H., Zhang, K., Easter, R. C., Tilmes, S., Fast, J. D., Liu, X., Yoon, J.-H., and Lamarque, J.-F.: The role of circulation features on black carbon transport into the Arctic in the Community Atmosphere Model version 5 (CAM5), J. Geophys. Res.-Atmos., 118, 4657–4669, https://doi.org/10.1002/jgrd.50411, 2013.*

*Wang, H., Rasch, P. J., Easter, R. C., Singh, B., Zhang, R., Ma, P.-L., Qian, Y., Ghan, S. J., and Beagley, N.: Using an explicit emission tagging method in global modeling of source-receptor relationships for black carbon in the Arctic: variations, sources, and transport pathways, J. Geophys. Res.-Atmos., 119, 12888– 12909, https://doi.org/10.1002/2014JD022297, 2014.*

Page 18, line 25: Please provide a reference for the assertion regarding expected future increases in Arctic shipping.

*Response: We have added references for expected future increases in Arctic shipping. The manuscript has been edited as follows: "Understanding the impacts of ship emissions on climate and air quality of the Arctic environment is challenging but important, given the likelihood of future increases in Arctic shipping (Corbett et al., 2010; Pizzolato et al., 2014; Winther et al., 2014)." (page 20, line 17)*

*Corbett, J. J., Lack, D. A., Winebrake, J. J., Harder, S., Silberman, J. A., and Gold, M.: Arctic shipping emissions inventories and future scenarios, Atmos. Chem. Phys., 10, 9689–9704, doi:10.5194/acp-10-9689-2010, 2010.*

*Pizzolato, L., Howell, S. E. L., Derksen, C., Dawson, J., and Copland, L.: Changing sea ice conditions and marine transportation activity in Canadian Arctic waters between 1990 and 2012, Climatic Change, 123, 161–173, doi:10.1007/s10584-013-1038-3, 2014.*

*Winther, M., Christensen, J. H., Plejdrup, M. S., Ravn, E. S., Eriksson, O. F., and Kristensen, H. O.: Emission inventories for ships in the arctic based on satellite sampled AIS data, Atmos. Environ., 91, 1–14, doi:10.1016/j.atmosenv.2014.03.006, 2014.*

Page 25, line 6: ".. transport of long-range pollutants.." Not sure what would be defined as a "long-range pollutant". Should this be "long-range transport of pollutants.."?

*Response:  This change has been made.*

Page 25, line 24: Since the IMPAACT project is not yet funded, it might be helpful to explicitly mention that IMPAACT is part of a broader activity aimed at reducing uncertainties in pollution processing during LRT to the Arctic under the PACES umbrella (https://pacesproject.org/about).

*Response: New text has been added: "IMPAACT is one effort of PACES which is a broader activity aimed at reducing uncertainties associated with pollution in the Arctic (https://pacesproject.org/about)." (page 27, line 15)*